# MULTI-ACTION SELF-IMPROVEMENT FOR NEURAL COMBINATORIAL OPTIMIZATION

**Laurin Luttmann**
Leuphana University Lüneburg
`laurin.luttmann@leuphana.de`

**Lin Xie**
Brandenburg University of Technology
`lin.xie@b-tu.de`

## ABSTRACT

Self-improvement has emerged as a state-of-the-art paradigm in Neural Combinatorial Optimization (NCO), where models iteratively refine their policies by generating and imitating high-quality solutions. Despite strong empirical performance, existing methods face key limitations. Training is computationally expensive, as policy updates require sampling numerous candidate solutions per instance to extract a single expert trajectory. More fundamentally, these approaches fail to exploit the structure of combinatorial problems involving the coordination of multiple agents, such as vehicles in min-max routing or machines in scheduling. By supervising on single-action trajectories, they fail to exploit agent-permutation symmetries, where distinct sequences of actions yield identical solutions, hindering generalization and the ability to learn coordinated behavior.

We address these challenges by extending self-improvement to operate over joint multi-agent actions. Our model architecture predicts complete agent-task assignments jointly at each decision step. To explicitly leverage symmetries, we employ a set-prediction loss, which supervises the policy on multiple expert assignments for any given state. This approach enhances sample efficiency and the model's ability to learn coordinated behavior. Furthermore, by generating multi-agent actions in parallel, it drastically accelerates the solution generation phase of the self-improvement loop. Empirically, we validate our method on several combinatorial problems, demonstrating consistent improvements in the quality of the final solution and a reduced generation latency compared to standard self-improvement.

## 1 INTRODUCTION

End-to-end constructive Neural Combinatorial Optimization (NCO) has emerged as a powerful framework for solving combinatorial optimization (CO) problems by casting them as sequential Markov Decision Processes (MDPs), where a neural policy constructs solutions step by step (Bengio et al., 2021). Reinforcement Learning (RL)-based methods have proven especially effective in this setting, enabling models to learn solution strategies through interaction rather than relying on pre-existing expert data (Bello et al., 2017; Kool et al., 2019; Kwon et al., 2020; Kim et al., 2022).

Despite their flexibility, traditional RL approaches in NCO face significant challenges. The sparse reward signals inherent to combinatorial optimization – where a meaningful objective is available only upon solution completion – have largely limited training to the REINFORCE algorithm (Williams, 1992) and variants (Kool et al., 2019; Kim et al., 2022; Kwon et al., 2020). Since these methods require backpropagation through the entire solution trajectory, the resulting high memory requirements and risk of gradient instability have encouraged a "heavy-encoder, light-decoder" architectural paradigm. In this setup, a static representation of the initial problem is generated once and used for all subsequent decisions. This becomes a critical bottleneck, as it fails to reflect the evolving problem state, often resulting in suboptimal decisions in the later stages of solution construction (Drakulic et al., 2023; Luo et al., 2024). Recently, self-improvement methods have offered a compelling solution to these issues (Pirnay & Grimm, 2024; Corsini et al., 2024). During training, these algorithms generate numerous solutions for a given problem instance, identify the best-performing trajectory, and use it as a pseudo-expert example for imitation learning. Because the policy is trained on individual state-action pairs along this expert trajectory, it enables the use of more powerful, decoder-only architectures that can dynamically re-encode the state at each step.

However, this next-token prediction approach of self-improvement faces a fundamental limitation in multi-agent CO problems, where multiple decision entities (agents) must plan and coordinate their actions to achieve a shared objective. These problems often exhibit agent-permutation symmetries, meaning that different permutations of agent-task assignments can yield identical solutions. For example, in vehicle routing, assigning driver 1 to location A and then driver 2 to location B leads to the same overall route as the reverse assignment order (Figure 1). However, by supervising the policy on a single "best" next action per state, self-improvement implicitly enforces an arbitrary agent order and treats the remaining symmetric choices as errors. This reduces sample efficiency and limits the model's ability to learn coordination in multi-agent settings, thus limiting generalization performance.

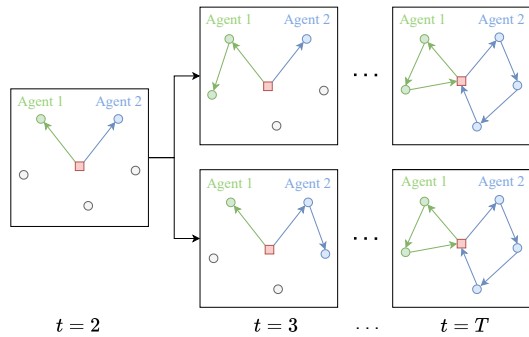

Figure 1: Example for agent-permutation symmetry. Both trajectories have the same solution.

To address these limitations, this paper introduces MACSIM – a **M**ulti-**AC**tion **S**elf-**I**mprovement **M**ethod. MACSIM extends the self-improvement paradigm by incorporating a multi-agent policy that predicts joint assignments for all agents in parallel at each decision step. To explicitly leverage problem symmetries, we employ a set-prediction loss, which allows the policy to learn from multiple equivalent expert assignments for a given state. This design promotes agent coordination, improves sample efficiency, and significantly accelerates solution generation. We demonstrate the effectiveness of MACSIM on several challenging combinatorial optimization tasks spanning both routing and scheduling domains. Our key contributions are as follows:

- We propose MACSIM, a novel learning paradigm for cooperative multi-agent combinatorial optimization problems that achieves better empirical results with lower solution generation latency compared to standard self-improvement methods.

- We develop a new solution generation scheme that models the entire joint-agent action space in a single forward pass – thus fostering coordination – and avoids conflicts between agents through an autoregressive sampling algorithm.

- We introduce a permutation-invariant surrogate loss function for imitation learning on expert multi-action trajectories, which stabilizes and accelerates the training process.

## 2 RELATED WORKS

The application of deep learning to CO problems was pioneered by the Pointer Network (Vinyals et al., 2015), which was trained via supervised learning to autoregressively construct solutions for the Traveling Salesman Problem (TSP). This paradigm was quickly adapted to RL, removing the dependency on optimal solutions as training data and thus improving scalability (Bello et al., 2017; Nazari et al., 2018). A major architectural advance came with transformer-based policies leveraging self-attention (Vaswani et al., 2017), leading to substantial performance gains (Kool et al., 2019).

A common strategy in these REINFORCE-based methods (Williams, 1992) is to apply a computationally expensive encoder once per problem instance and use the resulting embeddings in a lightweight, iterative decoder. While efficient, this "encode-once" approach has been shown to generalize poorly to out-of-distribution instances (Manchanda et al., 2022). In response, recent works have explored alternative strategies, such as re-encoding the state at each decoding step (Drakulic et al., 2023; Luo et al., 2024), learning a diverse set of policies to improve performance at test time (Grinsztajn et al., 2023; Hottung et al., 2025), or abandoning autoregressive decoding entirely in favor of heatmap-based (Joshi et al., 2019; Fu et al., 2021; Qiu et al., 2022; Ye et al., 2023) or diffusion-based methods (Sun & Yang, 2023; Li et al., 2023; 2024).

Self-improvement learning has recently emerged as a powerful training paradigm that bypasses the need for expert solutions (Corsini et al., 2024; Pirnay & Grimm, 2024). Inspired by elite sampling from cross-entropy methods (Boer et al., 2005), these approaches sample multiple solution trajec-

tories, identify the best-performing one as a "pseudo-expert" and train the policy to imitate this target via next-token prediction. However, this approach assumes a unique optimal action per step, which is often violated in CO problems that exhibit symmetries (Cappart et al., 2023). In such cases, forcing the model to follow a single sequence reduces solution diversity and impairs generalization.

Several lines of research have focused explicitly on exploiting symmetries. POMO (Kwon et al., 2020) and Sym-NCO (Kim et al., 2022) leverage the cyclic symmetry of the TSP by training on rollouts from all possible starting nodes and using their averaged returns to compute a low-variance baseline. More recently, DPN (Zheng et al., 2024) generalized this concept to multi-agent problems like the min-max VRP, where the order of agent routes is permutation-invariant. DPN samples multiple agent orderings and computes an agent-permutation-symmetric baseline, which reduces gradient variance and accelerates convergence by avoiding redundant learning.

We extend self-improvement to explicitly leverage these structural symmetries in multi-agent CO problems. In contrast to PARCO, which resolves conflicts post-hoc, and DPN, which reduces variance through permutation-invariant baselines, MACSIM generates coordinated and conflict-free assignments directly and learns with a set-based loss, leading to more robust symmetry-aware policies.

## 3 PRELIMINARIES

NCO typically involves a reformulation of the respective CO problem into a Markov Decision Process (MDP), where a solution $\tau$ to a problem instance $x$ is constructed over a finite horizon $t = 1, \ldots, T$. The current configuration of the problem at time step $t$ is encoded in the state $s_t \in \mathcal{S}$, which typically represents all information in a graph $\mathcal{G} = \{\mathcal{V}, \mathcal{E}, w\}$, with $N$ nodes $\mathcal{V}$, edges $\mathcal{E} \subseteq \mathcal{V} \times \mathcal{V}$ and edge weights $w : \mathcal{E} \to \mathbb{R}$ (Khalil et al., 2017). At each step, a policy $\pi_\theta$ with learnable weights $\theta$ selects an action $a_t \in \mathcal{V}$ based on $s_t$, and the transition function $\psi$ updates the state to $s_{t+1}$ (Khalil et al., 2017). Due to the sequential structure of MDPs, NCO commonly adopts autoregressive (AR) policies that generate actions step by step, conditioned on prior decisions. Classical approaches trained via REINFORCE typically encode only the initial state $s_0$ using a graph neural network, while the policy operates autoregressively from this encoding. In contrast, self-improvement methods enable training at the action level, allowing the policy to encode intermediate states $s_t$ and use decoder-only architectures defined as $P(\tau \mid x; \theta) = \prod_{t=1}^{T} \pi_\theta(a_t \mid s_t)$.

In this work, we specifically target CO problems that can be framed as a cooperative multi-agent MDP (MMDP or Markov Game) with $M$ agents sharing a common reward (Boutilier, 1996). Agents correspond to decision entities executing tasks, such as machines in scheduling problems or vehicles in routing. The state $s_t$ of the problem can be defined by a bipartite graph $\mathcal{G} = \{\mathcal{V}, \mathcal{M}, \mathcal{E}, w\}$, where $\mathcal{V}$ is the set of nodes (tasks), $\mathcal{M}$ is the set of agents, and edges $\mathcal{E} \subseteq \mathcal{M} \times \mathcal{V}$ denote feasible agent-task assignments, each associated with a cost $w : \mathcal{E} \to \mathbb{R}$.

At each step $t$, the policy $\pi_\theta$ maps the state $s_t$ to a bipartite matching $\mathbf{a}_t = \{a_t^k\}_{k=1}^{M}$ between agent and tasks. Each element $a_t^k = (m_k, v_k)$ denotes the assignment of task $v_k$ to agent $m_k$, where no two agents can be assigned to the same task. Given a matching $\mathbf{a}_t \in \mathcal{A}$, with $\mathcal{A}$ the space of valid matchings, the problem transitions from $s_t$ to $s_{t+1}$ according to a transition function $\psi : \mathcal{S} \times \mathcal{A} \to \mathcal{S}$, until a complete solution $\tau$ is obtained. The transition function is assumed order-invariant, meaning it depends only on the final matching $\mathbf{a}_t$, not on the order in which assignments are produced. Agents receive a shared reward observed only at terminal states, where the return $R(\tau, x)$ equals the negative value of the CO objective for the complete solution.

## 4 MULTI-ACTION SELF-IMPROVEMENT

### 4.1 MULTI-AGENT POLICY

The permutation invariance of the transition function with respect to joint agent-task assignments induces symmetries over agent orderings. Standard self-improvement typically ignores this structure by assuming a single best action per time-step. In contrast, our approach leverages these agent-permutation symmetries by learning a policy that generates complete joint agent-task assignments $\mathbf{a}_t$ instead of single next actions $a_t = (m, v)$. To this end, MACSIM utilizes a multi-agent policy that directly maps the current state $s_t$ to a joint agent-action assignment $\mathbf{a}_t$. This policy first encodes

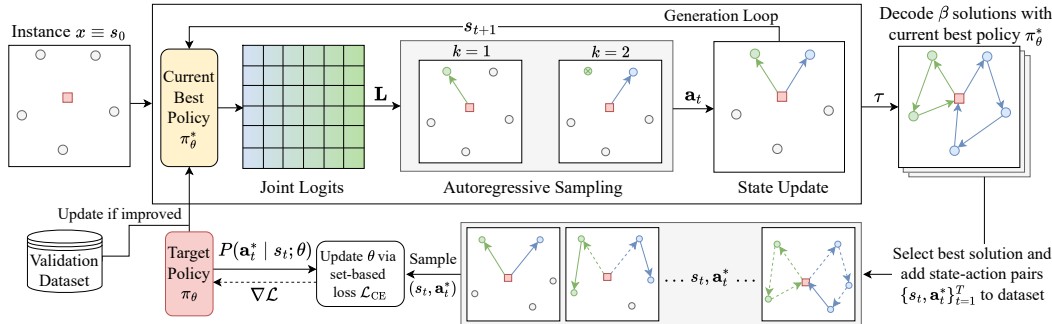

Figure 2: Overview of MACSIM. It generates joint agent-action logits in parallel through a multi-agent policy and autoregressively samples from them to generate complete agent-task assignments. The policy is used to sample $\beta$ solutions, where the best serves as training example. A set-based loss function is used to train the policy on pseudo-expert multi-agent actions for a given state.

the bipartite input graph $\mathcal{G}$ into agent embeddings $\mathbf{H}_{\mathcal{M}} \in \mathbb{R}^{M \times d}$ and task embeddings $\mathbf{H}_{\mathcal{V}} \in \mathbb{R}^{N \times d}$. Based on these embeddings, it computes a matrix of unnormalized logits $\mathbf{L} \in \mathbb{R}^{M \times N}$, where each entry $L_{m,v}$ represents the compatibility score for assigning task $v$ to agent $m$. For full architectural details of the policy network, we refer the reader to Appendix B.

While the idea of multi-agent policies in NCO is not novel, existing approaches either sample only once from the resulting joint-distribution (Liu et al., 2024), thus discarding learned inter-agent correlations as well as potential efficiency gains through multi-step prediction. Or, they assume independence across agents by normalizing $\mathbf{L}$ per agent and sampling independently from the resulting marginal distributions. However, this independence assumption does not hold, leading to suboptimal coordination and conflicts, where agents happen to select the same action. PARCO (Berto et al., 2024) resolves conflicts by a post-hoc resolution mechanism, prioritizing agents with larger log-probabilities while leaving others idle. Yet, since agents are modeled independently through different marginal distributions, effective coordination does not take place.

## 4.2 MULTI-ACTION GENERATION

To address this, we introduce an autoregressive sampling procedure, that utilizes the joint-logits $\mathbf{L}$ from the neural policy to sequentially sample agent-action pairs from their joint distribution without replacement. To this end, the probability of generating a specific sequence $\mathbf{a}_t$ is factorized using the chain rule of probability:

$$P(\mathbf{a}_t \mid \mathbf{L}) = \prod_{k=1}^{M} P\big(m_{t,k}, v_{t,k} \mid \mathbf{a}_t^{<k}, \mathbf{L}\big) \tag{1}$$

where $\mathbf{a}_t^{<k} = ((m_{t,1}, v_{t,1}), \ldots, (m_{t,k-1}, v_{t,k-1}))$ denotes the sequence of previous assignments in the construction of $\mathbf{a}_t$. This formulation casts the selection of each agent-task pair as a step in a sequence, where the choice at step $k$ is conditioned on all prior assignments. Specifically, let $\mathcal{M}_k$ and $\mathcal{V}_k$ be the sets of agents and tasks, respectively, available at step $k$, with $\mathcal{M}_1 = \mathcal{M}$ and $\mathcal{V}_1 = \mathcal{V}$. At each step $k \in \{1, \ldots, M\}$, the policy samples a feasible and available agent-task pair, $(m_k, v_k) \in \mathcal{E}_k \subseteq \mathcal{E}$, where $\mathcal{E}_k = \mathcal{E} \cap (\mathcal{M}_k \times \mathcal{V}_k)$, from a categorical distribution with probabilities proportional to their scores $L_{m_k, v_k}$:

$$P\big(a_t^k = (m_k, v_k) \mid \mathbf{a}_t^{<k}, \mathbf{L}\big) = \frac{\exp(L_{m_k, v_k})}{\sum_{(m', v') \in \mathcal{E}_k} \exp(L_{m', v'})}, \tag{2}$$

After sampling $a_t^k = (m_k, v_k)$, the sets for the next step are updated via $\mathcal{M}_{k+1} = \mathcal{M}_k \setminus \{m_k\}$ and $\mathcal{V}_{k+1} = \mathcal{V}_k \setminus \{v_k\}$ and the process continues until all agents are assigned to a task. We summarize this sampling process, which guarantees the generation of a valid matching $\mathbf{a}_t$, in Algorithm 1. The validity of the resulting distribution is formally stated as follows:

**Proposition 1.** *The function $P(\mathbf{a}_t \mid \mathbf{L})$ defined by the autoregressive process in Equations 1 and 2 is a valid probability distribution over the space of all possible ordered agent-task matchings.*[1]

---

[1]A proof for Proposition 1 can be found in Appendix A.

The injectivity constraint implies that earlier assignments reduce the set of available actions for subsequent agents, making the generation process order-sensitive, even though the final solution is permutation-invariant. Our autoregressive approach explicitly models this dependency: the joint softmax normalization dynamically integrates coordination, since the denominator depends on all remaining agent-task pairs. High logits for a pair $(m^*, v^*)$ implicitly reduce the probability of competing agents choosing the same task, allowing the policy to prioritize favorable assignments and avoid conflicts without relying on heuristics or post hoc conflict resolution, as PARCO does.

**Algorithm 1** Joint Action Sampling from $\mathbf{L}$

---

**Require:** Score matrix $\mathbf{L} \in \mathbb{R}^{M \times N}$, feasible pairs $\mathcal{E}$
**Ensure:** $\mathbf{a} = [(m_1, v_1), \ldots, (m_M, v_M)]$
1: $\mathbf{a} \leftarrow [\,]$        ▷ Initialize empty sequence
2: **for** $k = 1$ to $M$ **do**
3:     $P(m, v) \leftarrow \underset{(m,v) \in \mathcal{E}}{\mathrm{softmax}}(L_{m,v})$    ▷ Normalize
4:     $(m_k, v_k) \sim P$      ▷ Sample from joint dist.
5:     $\mathbf{a} \leftarrow \mathbf{a} \, \| \, (m_k, v_k)$        ▷ Append
6:     $\mathbf{L}_{m_k,:} \leftarrow -\infty; \quad \mathbf{L}_{:,v_k} \leftarrow -\infty$    ▷ Mask
7: **end for**
8: **return** $\mathbf{a}$

---

Substituting the definition of $P(\mathbf{a}_t \mid \mathbf{L})$ from Equation (1) into the decoder-only policy of Section 3 yields the following definition of our MACSIM policy:

$$P(\tau \mid s_t; \theta) = \prod_{t=1}^{T} \pi_\theta(\mathbf{L} \mid s_t) \prod_{k=1}^{M} P(a_t^k \mid \mathbf{a}_t^{<k}, \mathbf{L}) \tag{3}$$

In this formulation, the joint logits $\mathbf{L}$ are computed only once by the computationally expensive neural policy $\pi_\theta$, after which $M$ actions are generated using a fast autoregressive sampling procedure. As a result, MACSIM can construct solutions significantly faster than fully autoregressive models, as we demonstrate in the experimental section of this paper.

## 4.3 SKIP TOKEN

The generative model defined by Equation (3) requires each agent to select an action at every decision step. However, enforcing task assignments for all agents at every step can be suboptimal, particularly in problems with strong inter-agent dependencies. In job-shop scheduling, for instance, the set of available jobs for a machine likely changes as other machines schedule operations. In such cases, it may be beneficial for an agent to wait until a more favorable task becomes available.

Therefore, we introduce a dummy action, referred to as the skip token. The skip token is a transient action that can be chosen by any agent at any time, allowing them to wait until the next decision step without modifying the current solution. To encourage efficient solution construction, each use of the skip token incurs a small penalty added to the objective value. Empirically, we find that annealing this penalty toward zero during training yields the best performance. This strategy encourages the policy to increasingly prioritize generating high-quality solutions as training progresses. We examine the effect of the skip token in Table 3 and provide more details and analyses in Appendix D.3.

Technically, the skip token is implemented as a learnable embedding $\mathbf{h}^{\mathrm{skip}} \in \mathbb{R}^d$ which is added to the set of all task embeddings $\mathbf{H}_\mathcal{V}$. Unlike other actions, the skip token can be selected by multiple agents, with the only restriction that at least one agent selects an actual task.

## 4.4 POLICY LEARNING

We train MACSIM with a two-stage self-improvement framework, similar to Pirnay & Grimm (2024) and Corsini et al. (2024). In the first stage, the current best policy $\pi_\theta^*$ generates $\beta \gg 1$ solutions for each problem instance $x$. We select the best solution with respect to the objective value plus the penalty term for skip token usage, and add the state-action pairs $((s_1, \mathbf{a}_1^*), \ldots, (s_T, \mathbf{a}_T^*))$ corresponding to the solution $\tau^*$ to the training dataset. In the second stage, the policy network $\pi_\theta$ is updated via imitation learning on these pseudo-expert trajectories.

Unlike prior work that predicts a single "token" from a partial solution, we aim to maximize the likelihood of the policy producing the entire expert multi-agent action $\mathbf{a}_t^*$ given the state $s_t$. The corresponding negative log-likelihood (NLL) under our generative process of Equation (1) is:

$$\mathcal{L}_{\mathrm{ML}} = -\sum_{k=1}^{M} \log P(m_k, v_k \mid \mathbf{a}_t^{<k}) = -\sum_{k=1}^{M} \log \left( \frac{\exp(L_{m_k, v_k})}{\sum_{(m', v') \in \mathcal{E}_k} \exp(L_{m', v'})} \right). \tag{4}$$

However, direct optimization with $\mathcal{L}_{\text{ML}}$ is problematic. As detailed in Appendix E, its primary flaw is conflicting gradient signals: each term in the sum assumes one assignment is correct at a given step and consequently penalizes all others, including those that occur later in the expert assignments.

A first step to mitigate this issue is to utilize the agent order from the expert assignment. By assuming a fixed agent order $\mathbf{m}^* = (m_1, \ldots, m_M)$ the choice of agent at step $k$ becomes deterministic. Thus, $P(m_k \mid \mathbf{a}_t^{<k}) = 1$ and the probability $P(v_k, m_k \mid \mathbf{a}_t^{<k})$ can be factorized as $P(v_k \mid m_k, \mathbf{a}_t^{<k})$, transforming the generative process into a Plackett-Luce (PL) model (Volkovs & Zemel, 2012). Its NLL computes the loss over the marginal distributions for each agent, preventing gradient conflicts:

$$\mathcal{L}_{\text{PL}} = -\sum_{k=1}^{M} \log P(v_k \mid m_k, \mathbf{a}_t^{<k}) = -\sum_{k=1}^{M} \log \frac{\exp(L_{m_k, v_k})}{\sum_{v' \in \mathcal{V}_k} \exp(L_{m_k, v'})}. \tag{5}$$

While this improves gradient stability, the loss remains sensitive to the permutation of the expert sequence. However, every permutation of the assignments in $\mathbf{a}^*$ yields the same solution, which should be reflected in the loss function. Ideally, this would be achieved by averaging the Plackett-Luce loss over all possible $M!$ agent orderings, which is computationally intractable. Therefore, we employ a *surrogate loss* by relaxing the sequential dependence assumption for the loss calculation and treating each agent-task pair in an expert matching as an independent supervised instance with conditional probability $P(v \mid m)$. This is justified because the expert data consist only of valid matchings, making it unnecessary to enforce the injectivity constraint within the loss itself. This is analogous to bipartite matching in object detection (Carion et al., 2020), where the assignment between predictions and ground-truth objects is first established algorithmically, and losses are then computed independently for each matched pair. The resulting surrogate is defined as the cross-entropy (CE) loss summed over all agents:

$$\mathcal{L}_{\text{CE}} = -\sum_{k=1}^{M} \log P(v_k \mid m_k) = -\sum_{k=1}^{M} \log \frac{\exp(L_{m_k, v_k})}{\sum_{v' \in \mathcal{V}} \exp(L_{m_k, v'})}. \tag{6}$$

This surrogate is permutation-invariant, computationally efficient, and provides a more robust training signal as we will validate empirically in the experimental section. Moreover, we provide a thorough analysis of $\mathcal{L}_{\text{CE}}$ in Appendices E and F.

## 5 EXPERIMENTS

We assess the effectiveness of MACSIM on representative multi-agent CO problems, spanning routing and scheduling domains. Specifically, we evaluate MACSIM on two challenging scheduling problems – the flexible job shop scheduling problem (FJSP) and the flexible flow shop problem (FFSP) – as well as a common routing problem, the heterogeneous capacitated vehicle routing problem (HCVRP). We compare MACSIM with common and SOTA solvers for the respective CO problems and the self-improvement method (**SLIM**) of Corsini et al. (2024); Pirnay & Grimm (2024).[2]

### 5.1 PROBLEMS

**Flexible Job Shop Scheduling Problem.**  The FJSP is concerned with scheduling $N$ jobs on $M$ machines (agents). Each job consists of a sequence of operations that must be executed in a fixed order. Unlike the classical job shop problem, each operation in FJSP can be processed by a subset of eligible machines $\mathcal{M}_k \subseteq \mathcal{M}$, with machine-dependent processing times, resulting in a combined routing (assigning operations to machines) and sequencing problem (ordering operations on each machine). The common objective is to minimize the makespan of the resulting schedule. The formal mathematical model and the MMDP formulation of the FJSP are presented in Appendix C.1.

To evaluate the performance on the FJSP, we compare MACSIM against several baselines. First, we include OR-Tools, a state-of-the-art CP-SAT solver widely applied in scheduling (Col & Teppan, 2019). We also benchmark against classical priority dispatching rules – FIFO (Firt In First Out), MOR (Most Operations Remaining), and MWKR (Most Work Remaining) – commonly used in manufacturing scheduling (Montazeri & Wassenhove, 1990). Lastly, we consider learning-based approaches, including a graph neural network trained via Proximal Policy Optimization (PPO) proposed by Song et al. (2022), and the dual attention model DANIEL (Wang et al., 2023).

---

[2]Source code is available at https://github.com/LTluttmann/macsim

**Flexible Flow Shop Scheduling Problem.** The FFSP involves scheduling $N$ jobs that must pass through $S$ sequential processing stages. The key challenge lies in the flexibility at each stage, which contains $M$ parallel machines; a job can be processed by any available machine within a stage. The objective is to determine the assignment and sequence of jobs on machines to minimize the makespan. A detailed problem formulation for the FFSP is provided in Appendix C.2.

We compare MACSIM against traditional baselines – Gurobi (Gurobi Optimization, LLC, 2025), a Genetic Algorithm (Hejazi & Saghafian, 2005), and Particle Swarm Optimization (Singh & Mahapatra, 2012) – as well as neural baselines: MatNet (Kwon et al., 2021), PolyNet (Hottung et al., 2025), and PARCO (Berto et al., 2024).

**Min-Max Heterogeneous Capacitated Vehicle Routing Problem.** The HCVRP is a challenging extension of the classical CVRP, designed to capture more realistic logistics and transportation scenarios. In HCVRP, a fleet of heterogeneous vehicles, each with distinct capacities and travel costs, is responsible for serving a set of customer demands. The objective differs from standard CVRP: instead of minimizing the total cost or distance, the goal is to minimize the maximum route length (or workload) among all vehicles, ensuring a balanced distribution of effort across the fleet. A detailed mathematical formulation for the min-max HCVRP is provided in Appendix C.3.

For the HCVRP, we employ two well-known heuristic baselines: Simulated Annealing (İlhan, 2021) and SISR (Christiaens & Berghe, 2020), a state-of-the-art heuristic for solving the CVRP and variants. Neural baselines involve Equity-Transformer (ET) (Son et al., 2024), $DRL_{LI}$ (Li et al., 2022), 2D-Ptr (Liu et al., 2024), and DPN (Zheng et al., 2024).

## 5.2 EXPERIMENTAL RESULTS

We present the main empirical results in Table 1, which reports test-set performance of the evaluated methods in terms of average objective values (Obj.), gaps to the best-known solutions, and average inference latency for a single instance. For neural baselines, we report performance under both greedy (*g.*) and sampling (*s.*) decoding, with the latter evaluated using 1,280 sampled solutions (more details on the experimental setup can be found in Appendix H). MACSIM consistently outperforms neural baselines across all problem types and sizes, surpasses all methods on FFSP, and substantially reduces the gap to OR-Tools on FJSP, even outperforming it on $20 \times 5$ instances.

Compared to SLIM, the advantage of MACSIM increases with the number of agents, as shown in Figure 3b for the HCVRP and FFSP. In addition, MACSIM significantly reduces inference time compared to SLIM. For FFSP $50 \times 4$ instances, SLIM requires nearly ten times longer than MACSIM to generate a solution. This difference grows with the number of agents, as shown in Figure 3c, which illustrates the number of forward passes needed by each policy to construct a solution and the resulting inference times. MACSIM requires only a fraction of the construction steps used by SLIM, and this advantage further increases with the number of agents due to its multi-agent policy.

While MACSIM is not necessarily the fastest among the neural solvers evaluated here, slower inference times compared to other methods can be attributed to the fact that, like SLIM, MACSIM performs step-wise re-encoding of the problem state. As discussed in previous work, this enables better generalization to out-of-distribution instances (e.g., Luo et al. (2024)). Table 2 validates this by evaluating models trained on the small FJSP instances of Table 1 on larger instances, where MACSIM outperforms all neural baselines and even OR-Tools on two out of three instance types.

We also conduct an ablation study to assess the impact of MACSIM's components. First, Figure 3a evaluates MACSIM with different sequence generation modes. While "MACSIM-full" refers to the generative model defined by Equation (3) with the sequential sampling algorithm described in Algorithm 1, "MACSIM-random-order" and "MACSIM-fixed-order" denote variants where agents act in random or fixed order, respectively, given logits **L**. The results in Figure 3a and table 3 highlight the effectiveness of the proposed sampling procedure, particularly on instances with many agents. Moreover, Table 3 confirms the importance of the skip token, which proves especially beneficial in larger instances where coordination among agents is more challenging. Although the skip token increases generation latency, it substantially improves solution quality. A more detailed analysis of the skip token and its penalty is provided in Appendix D where we provide more experiments.

Table 1: Test set performance of MACSIM and baselines on FJSP, FFSP, and HCVRP. Best obj. values found by any solver are shown in bold; grey backgrounds indicate the best neural solver.

**FJSP**

| $N \times M$ | $10 \times 5$ | | | $20 \times 5$ | | | $15 \times 10$ | | |
|---|---|---|---|---|---|---|---|---|---|
| Metric | Obj. | Gap | Time (s.) | Obj. | Gap | Time (s.) | Obj. | Gap | Time (s.) |
| OR-Tools | **96.32** | 0.00% | 1597 | 188.15 | 0.03% | 1800 | **143.53** | 0.00% | 1724 |
| FIFO | 119.4 | 23.96% | 0.16 | 216.08 | 14.88% | 0.32 | 184.55 | 28.58% | 0.51 |
| MOR | 115.38 | 19.79% | 0.16 | 214.16 | 13.85% | 0.32 | 173.15 | 20.64% | 0.51 |
| MWKR | 113.23 | 17.56% | 0.16 | 209.78 | 11.53% | 0.32 | 171.25 | 19.31% | 0.50 |
| PPO (*g.*) | 111.67 | 15.94% | 0.45 | 211.22 | 12.29% | 1.43 | 166.92 | 16.30% | 1.35 |
| DANIEL (*g.*) | 106.71 | 10.79% | 0.45 | 197.56 | 5.03% | 0.94 | 161.28 | 12.37% | 1.43 |
| SLIM (*g.*) | 103.85 | 7.82% | 0.91 | 194.37 | 3.33% | 1.18 | 154.32 | 7.62% | 2.74 |
| MACSIM (*g.*) | 102.21 | 6.12% | 0.44 | 191.08 | 1.58% | 0.76 | 149.84 | 4.40% | 1.32 |
| PPO (*s.*) | 105.59 | 9.62% | 1.11 | 207.53 | 10.33% | 2.36 | 160.86 | 12.07% | 6.42 |
| DANIEL (*s.*) | 101.67 | 5.55% | 0.74 | 192.78 | 2.49% | 1.87 | 153.22 | 6.75% | 6.10 |
| SLIM (*s.*) | 98.74 | 2.51% | 2.32 | 189.08 | 0.52% | 6.91 | 149.02 | 3.82% | 20.08 |
| MACSIM (*s.*) | 97.64 | 1.37% | 0.86 | **188.10** | 0.00% | 2.28 | 145.95 | 1.69% | 6.19 |

**FFSP**

| $N \times M_i \times S$ | $20 \times 4 \times 3$ | | | $50 \times 4 \times 3$ | | | $100 \times 4 \times 3$ | | |
|---|---|---|---|---|---|---|---|---|---|
| Metric | Obj. | Gap | Time (s.) | Obj. | Gap | Time (s.) | Obj. | Gap | Time (s.) |
| Gurobi (600s) | 31.61 | 31.93% | 600 | - | - | 600 | - | - | 600 |
| Genetic Algorithm | 31.15 | 30.01% | 21.05 | 56.92 | 19.23% | 44.82 | 99.25 | 13.79% | 89.20 |
| Particle Swarm | 29.10 | 21.45% | 46.17 | 55.10 | 15.42% | 82.46 | 97.30 | 11.56% | 154 |
| MatNet (*g.*) | 27.26 | 13.77% | 1.22 | 51.52 | 7.92% | 2.17 | 91.58 | 5.00% | 4.97 |
| PolyNet (*g.*) | 26.71 | 11.48% | 1.69 | 51.01 | 6.85% | 2.45 | 91.22 | 4.59% | 5.21 |
| PARCO (*g.*) | 26.31 | 9.81% | 0.26 | 51.19 | 7.23% | 0.52 | 91.29 | 4.67% | 0.89 |
| SLIM (*g.*) | 26.18 | 9.27% | 0.86 | 50.01 | 4.75% | 3.36 | 91.97 | 5.45% | 5.14 |
| MACSIM (*g.*) | 25.75 | 7.47% | 0.28 | 49.36 | 3.39% | 0.43 | 89.88 | 1.86% | 0.96 |
| MatNet (*s.*) | 25.43 | 6.14% | 3.88 | 49.68 | 4.06% | 8.91 | 89.72 | 2.87% | 18.00 |
| PolyNet (*s.*) | 24.98 | 4.26% | 5.04 | 49.23 | 3.12% | 9.24 | 89.21 | 2.28% | 19.29 |
| PARCO (*s.*) | 24.78 | 3.42% | 0.99 | 49.27 | 3.20% | 1.97 | 89.46 | 2.57% | 4.04 |
| SLIM (*s.*) | 24.19 | 0.96% | 1.55 | 48.13 | 0.82% | 10.21 | 89.50 | 2.61% | 19.01 |
| MACSIM (*s.*) | **23.96** | 0.00% | 0.49 | **47.74** | 0.00% | 0.91 | **87.22** | 0.00% | 3.60 |

**HCVRP**

| $N \times M$ | $60 \times 3$ | | | $80 \times 3$ | | | $100 \times 3$ | | |
|---|---|---|---|---|---|---|---|---|---|
| Metric | Obj. | Gap | Time (s.) | Obj. | Gap | Time (s.) | Obj. | Gap | Time (s.) |
| SISRs | **6.57** | 0.00% | 478 | **8.52** | 0.00% | 750 | **10.29** | 0.00% | 1084 |
| Simulated Annealing | 7.04 | 7.15% | 382 | 9.17 | 7.63% | 561 | 11.13 | 8.16% | 765 |
| ET (*g.*) | 7.58 | 15.37% | 0.28 | 9.76 | 14.55% | 0.38 | 11.74 | 14.09% | 0.45 |
| DPN (*g.*) | 7.46 | 13.54% | 0.28 | 9.66 | 13.38% | 0.40 | 11.48 | 11.56% | 0.46 |
| $\text{DRL}_{Li}$ (*g.*) | 7.43 | 13.09% | 0.34 | 9.64 | 13.15% | 0.46 | 11.44 | 11.18% | 0.58 |
| 2D-Ptr (*g.*) | 7.20 | 9.59% | 0.20 | 9.24 | 8.45% | 0.27 | 11.12 | 8.07% | 0.31 |
| SLIM (*g.*) | 7.19 | 9.44% | 0.63 | 9.25 | 8.57% | 0.87 | 11.10 | 7.87% | 1.04 |
| MACSIM (*g.*) | 7.15 | 8.83% | 0.35 | 9.15 | 7.39% | 0.43 | 11.02 | 7.09% | 0.78 |
| ET (*s.*) | 7.14 | 8.68% | 0.52 | 9.19 | 7.86% | 0.66 | 11.20 | 8.84% | 1.02 |
| DPN (*s.*) | 7.03 | 7.00% | 0.55 | 9.16 | 7.51% | 0.71 | 11.03 | 7.19% | 1.08 |
| $\text{DRL}_{Li}$ (*s.*) | 6.97 | 6.09% | 0.73 | 9.10 | 6.81% | 1.10 | 10.90 | 5.93% | 1.48 |
| 2D-Ptr (*s.*) | 6.82 | 3.81% | 0.32 | 8.85 | 3.87% | 0.44 | 10.71 | 4.08% | 0.55 |
| SLIM (*s.*) | 6.88 | 4.75% | 2.40 | 8.92 | 4.69% | 3.31 | 10.81 | 5.05% | 4.09 |
| MACSIM (*s.*) | 6.76 | 2.89% | 1.65 | 8.78 | 3.05% | 2.29 | 10.67 | 3.69% | 2.76 |

Figure 3: **Left**: Validation performance (makespan) on FJSP $10 \times 5$ instances during training under different sequence generation strategies. **Middle**: Average performance gap of SLIM relative to MACSIM across varying numbers of agents. **Right**: Inference efficiency comparison, reporting construction steps (lines, right axis) and inference time to generate a solution (bars, left axis).

Table 2: Generalization performance of MACSIM and baseline solvers on larger FJSP instance distributions not seen during training.

| | **FJSP** | | | | | | | | |
|---|---|---|---|---|---|---|---|---|---|
| $N \times M$ | | $20 \times 10$ | | | $30 \times 10$ | | | $40 \times 10$ | |
| Metric | Obj. | Gap | Time (s.) | Obj. | Gap | Time (s.) | Obj. | Gap | Time (s.) |
| OR-Tools | 195.98 | 3.14% | 1800 | **274.67** | 0.00% | 1800 | 365.96 | 0.08% | 1800 |
| PPO (g.) | 215.78 | 13.56% | 1.91 | 312.59 | 13.81% | 2.86 | 416.18 | 13.81% | 3.82 |
| DANIEL (g.) | 198.50 | 4.46% | 1.85 | 281.49 | 2.48% | 2.76 | 371.45 | 1.58% | 3.81 |
| SLIM (g.) | 195.89 | 3.09% | 3.11 | 281.87 | 2.62% | 4.57 | 374.13 | 2.31% | 6.03 |
| MACSIM (g.) | 192.15 | 1.12% | 1.19 | 276.01 | 0.49% | 1.71 | 365.87 | 0.05% | 2.27 |
| PPO (s.) | 214.81 | 13.05% | 6.23 | 308.55 | 12.33% | 12.79 | 410.76 | 12.33% | 24.54 |
| DANIEL (s.) | 193.91 | 2.05% | 6.35 | 279.20 | 1.65% | 12.37 | 370.08 | 1.21% | 21.09 |
| SLIM (s.) | 194.19 | 2.19% | 28.15 | 281.42 | 2.46% | 69.97 | 373.70 | 2.20% | 139.30 |
| MACSIM (s.) | **190.02** | 0.00% | 6.79 | 275.48 | 0.29% | 14.12 | **365.67** | 0.00% | 27.13 |

Table 3: Ablation study on MACSIM components

| | **FJSP** | | | |
|---|---|---|---|---|
| $N \times M$ | | $10 \times 5$ | | $15 \times 10$ |
| Metric | Obj. | Time | Obj. | Time |
| MACSIM-full (s.) | **97.64** | 0.86 | **145.95** | 6.19 |
| w/o AR-sampling   fixed (s.) | 98.97 | 0.76 | 155.13 | 3.97 |
| random (s.) | 98.66 | 0,78 | 150.72 | 4.15 |
| w/o skip-token (s.) | 98.69 | **0.75** | 159.65 | **3.89** |

Table 4: Comparison of Loss Functions

| | **FJSP** | | **FFSP** | |
|---|---|---|---|---|
| $N \times M$ | $10 \times 5$ | $20 \times 5$ | $20 \times 12$ | $50 \times 12$ |
| Metric | Obj. | Obj. | Obj. | Obj. |
| $\mathcal{L}_{\text{SA}}$ (s.) | 98.81 | 189.18 | 24.29 | 49.58 |
| $\mathcal{L}_{\text{ML}}$ (s.) | 98.13 | 188.97 | 24.15 | 49.02 |
| $\mathcal{L}_{\text{PL}}$ (s.) | 97.99 | 188.81 | 24.08 | 47.90 |
| $\mathcal{L}_{\text{CE}}$ (s.) | **97.64** | **188.10** | **23.96** | **47.74** |

Finally, Table 4 compares the loss functions from Section 4.4, along with the single-agent cross-entropy loss $\mathcal{L}_{\text{SA}}$ defined in Appendix E.1. Our surrogate set-based cross-entropy loss consistently achieves the best performance on both FJSP and FFSP.

## 6 CONCLUSION

We presented MACSIM, a novel self-improvement framework tailored for multi-agent CO problems. Instead of predicting a single next token, MACSIM learns a multi-agent policy that emits joint agent–task logits in one forward pass and then samples a valid matching autoregressively without replacement. This design captures inter-agent dependencies, prevents conflicts by construction, and amortizes policy computation over multiple assignments, drastically accelerating solution generation. A second key ingredient is a permutation-invariant, set-based learning objective. By supervising on the set of expert agent–task pairs rather than a single agent's next step, MACSIM explicitly exploits agent-permutation symmetries, avoids gradient conflicts inherent to single-action imitation, and promotes coordinated behaviors. A skip token combined with an annealed penalty further enables agents to defer actions when beneficial while keeping construction efficient.

Empirically, MACSIM achieves state-of-the-art results on challenging scheduling and routing benchmarks, consistently improving solution quality. By leveraging the structural symmetries of multi-agent problems, our approach enhances both the performance and scalability of neural solvers, making self-improvement practical for complex, real-world optimization tasks. Overall, MACSIM advances self-improvement for multi-agent CO by unifying joint-action modeling and symmetry-aware learning, yielding a more coordinated, generalizable, and practical solver.

**Limitations and Future Work.** Although MACSIM substantially improves inference and training time (see Figure 6 for training times) compared to standard self-improvement, its reliance on step-wise encoding results in slower generation than some lightweight neural solvers, which may hinder applicability in latency-critical settings. Future work will therefore focus on further reducing generation latency through more efficient policy architectures. A promising direction is to encode and cache the initial problem state once, and then incrementally update embeddings as the state evolves, rather than re-encoding the entire graph at each step. This strategy could retain MACSIM's ability to leverage multi-agent symmetries while reducing both memory and computational overhead.

## REPRODUCIBILITY STATEMENT

To facilitate reproducibility of our results, we provide complete access to our implementation, model weights, and training configurations. The source code is available at https://github.com/LTluttmann/macsim, and the trained model weights along with all configuration files used in our experiments can be found at https://osf.io/5z2aj/?view_only=783b0bb138e64431a681fd36452ea710. Detailed descriptions of model architectures, training procedures, and experimental setups are provided in the main paper and appendix. These resources together ensure that our experiments can be independently reproduced.

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

## A    PROOF OF PROPOSITION 1

**Proposition 1.** *The probability $P(\mathbf{a}_t \mid \mathbf{L})$ of generating a specific sequence of agent-task assignments defined by*

$$P(\mathbf{a}_t \mid \mathbf{L}) = \prod_{k=1}^{M} \frac{\exp(L_{m_k,v_k})}{\sum_{(m',v' \in \mathcal{E}_k)} \exp(L_{m',v'})} \tag{7}$$

*is a valid probability distribution over the space of all possible ordered agent-task matchings.*

*Proof.* To establish that $P(\mathbf{a} \mid \mathbf{L})$ is a valid probability distribution over the space $\mathcal{A}_{\text{seq}}$ of all ordered sequences of $M$ agent-task assignments, we demonstrate that it satisfies two axioms: (i) non-negativity and (ii) normalization.

**Non-negativity.**    Each term in Equation (7) is mapped into the probability simplex via the softmax function $f : \mathbb{R}^{\mathcal{E}_k} \to \Delta(|\mathcal{E}_k|)$ applied to the set of available agent-task pairs $\mathcal{E}_k = \mathcal{E} \cap (\mathcal{M}_k \times \mathcal{V}_k)$. By definition of the probability simplex, the softmax yields non-negative values that sum to one. Thus, each conditional probability $P(a_k \mid \mathbf{a}^{<k}, \mathbf{L})$ is non-negative, and the total probability $P(\mathbf{a} \mid \mathbf{L})$, as a product of non-negative terms, is also non-negative for any $\mathbf{a} \in \mathcal{A}_{\text{seq}}$, satisfying the non-negativity axiom.

**Normalization.**    To verify normalization, we demonstrate that the sum of probabilities over the entire sample space is unity. We expand this sum by expressing it as a nested sum over all possible choices at each step of the sequence generation. Using the notation $\mathcal{E}(\mathbf{a}^{<k})$ for the set of available choices at step $k$ given the history $\mathbf{a}^{<k}$ we have:

$$\sum_{\mathbf{a} \in \mathcal{A}_{\text{seq}}} P(\mathbf{a} \mid \mathbf{L}) = \sum_{\mathbf{a} \in \mathcal{A}_{\text{seq}}} \prod_{k=1}^{M} P(a_k \mid \mathbf{a}^{<k}, \mathbf{L})$$

$$= \sum_{a_1 \in \mathcal{E}} \sum_{a_2 \in \mathcal{E}(a_1)} \cdots \sum_{a_M \in \mathcal{E}(\mathbf{a}^{<M})} \prod_{k=1}^{M} P(a_k \mid \mathbf{a}^{<k}, \mathbf{L})$$

$$= \sum_{a_1 \in \mathcal{E}} P(a_1 \mid \mathbf{L}) \sum_{a_2 \in \mathcal{E}(a_1)} P(a_2 \mid a_1, \mathbf{L}) \cdots \sum_{a_M \in \mathcal{E}(\mathbf{a}^{<M})} P(a_M \mid \mathbf{a}^{<M}, \mathbf{L})$$

Consider the sum over all possible assignments for any given history $\mathbf{a}^{<k}$. By substituting the definition from Equation 2, we have:

$$\sum_{a_k \in \mathcal{E}(\mathbf{a}^{<k})} P(a_k = (m_k, v_k) \mid \mathbf{a}^{<k}, \mathbf{L}) = \sum_{(m_k, v_k) \in \mathcal{E}(\mathbf{a}^{<k})} \frac{\exp(L_{m_k, v_k})}{\sum_{(m',v') \in \mathcal{E}(\mathbf{a}^{<k})} \exp(L_{m',v'})}$$

$$= \frac{\sum_{(m_k, v_k) \in \mathcal{E}(\mathbf{a}^{<k})} \exp(L_{m_k, v_k})}{\sum_{(m',v') \in \mathcal{E}(\mathbf{a}^{<k})} \exp(L_{m',v'})}$$

$$= 1$$

Since $\sum_{a_k \in \mathcal{E}(\mathbf{a}^{<k})} P(a_k \mid \mathbf{a}^{<k}, \mathbf{L}) = 1$ for any $1 \le k \le M$, the marginalized sum collapses:

$$\sum_{\mathbf{a} \in \mathcal{A}_{\text{seq}}} P(\mathbf{a} \mid \mathbf{L}) = 1 \times \ldots \times 1 = 1.$$

Thus, the normalization property is satisfied. Since both axioms hold, $P(\mathbf{a} \mid \mathbf{L})$ is a valid probability distribution. $\square$

## B MACSIM POLICY

As defined in Section 3, the problems tackled by MACSIM can be formulated as MMDPs, whose states are bipartite graphs consisting of agent and task nodes. MACSIM therefore requires an architectural policy backbone, that is capable of encoding such graph structure effectively. Our policy is strongly motivated by related works of Kwon et al. (2021); Luttmann & Xie (2024; 2025), who define such architectures in the realm of NCO.

First, the policy employed in our paper projects the different node-types (agents and tasks) from their distinct feature spaces into a mutual embedding space of dimensionality $d$ using type-specific transformations $\mathbf{W}_{\epsilon_i}$ for node $i$ of type $\epsilon_i$. The features used to represent agents and tasks for the respective problems can be found in Appendix C.

Given the initial embeddings $\mathbf{H}^0_{\mathcal{M}}$ and $\mathbf{H}^0_{\mathcal{V}}$ for agents and tasks, respectively, we use several layers of self- and cross-attention to enable message passing between all nodes in the graph. While self-attention is applied independently to agent and task embeddings following the Transformer architecture Vaswani et al. (2017), cross-attention allows message passing between agent and task nodes.

Formally, to perform cross-attention we compute a matrix of attention scores $\mathbf{A}$ using agent embeddings as queries $\mathbf{Q}$ and task embeddings as keys $\mathbf{K}$:

$$\mathbf{A} = \frac{\mathbf{Q}\mathbf{K}^\top}{\sqrt{d_k}}$$

where

$$\mathbf{Q} = \mathbf{W}^{\mathbf{Q}}\mathbf{H}^{l-1}_{\mathcal{M}}, \qquad \mathbf{K} = \mathbf{W}^{\mathbf{K}}\mathbf{H}^{l-1}_{\mathcal{V}}$$

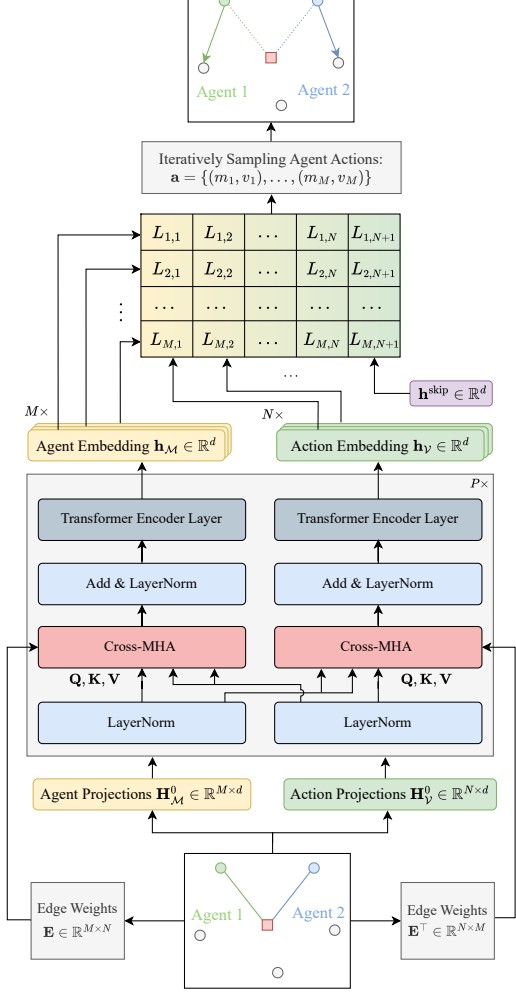

Figure 4: MACSIM Policy

and $\mathbf{W}^{\mathbf{Q}}$ and $\mathbf{W}^{\mathbf{K}} \in \mathbb{R}^{d_k \times d}$ are weight matrices learned per attention head[3] and $d_k$ is the per-head embedding dimension. The resulting attention scores $\mathbf{A} \in \mathbb{R}^{M \times N}$ can be interpreted as the (learned) compatibility of an agent $m$ and a task $v$. Similar to MatNet Kwon et al. (2021), we fuse these learned attention scores with the edge weights $w : \mathcal{E} \to \mathbb{R}$. In FFSP, for instance, the weight $w$ of an edge connecting an agent and a task node corresponds to the duration of the respective job on that specific machine. Formally, we concatenate the attention score and the matrix of edge weights $\mathbf{E}$ and feed the resulting score vector through a multi-layer perceptron $\mathrm{MLP} : \mathbb{R}^{M \times N \times 2} \to \mathbb{R}^{M \times N}$, with a single hidden layer comprising of $d$ units and GELU activation function Hendrycks & Gimpel (2016). Further, we pass the transpose of the attention scores and of the supply matrix $\mathbf{A}^\top$, $\mathbf{E}^\top \in \mathbb{R}^{N \times M}$ through a second MLP to obtain the reverse compatibility $A_{\mathcal{V} \to \mathcal{M}}$ of tasks $v$ on agents $m$:

$$\mathbf{A}_{\mathcal{M} \to \mathcal{V}} = \mathrm{MLP}_{\mathcal{M}}\big([\mathbf{A}||\mathbf{E}]\big), \quad \mathbf{A}_{\mathcal{V} \to \mathcal{M}} = \mathrm{MLP}_{\mathcal{V}}\big([\mathbf{A}^\top||\mathbf{E}^\top]\big), \tag{8}$$

The resulting attention scores are then used to compute the embeddings for the nodes of the respective type:

$$\mathbf{H}'_{\mathcal{M}} = \mathrm{softmax}(\mathbf{A}_{\mathcal{M} \to \mathcal{V}})\mathbf{V}_{\mathcal{V}}, \quad \mathbf{V}_{\mathcal{V}} = \mathbf{W}^{\mathbf{V}}_{\mathcal{V}}\mathbf{H}^{l-1}_{\mathcal{V}} \tag{9}$$

$$\mathbf{H}'_{\mathcal{V}} = \mathrm{softmax}(\mathbf{A}_{\mathcal{V} \to \mathcal{M}})\mathbf{V}_{\mathcal{M}}, \quad \mathbf{V}_{\mathcal{M}} = \mathbf{W}^{\mathbf{V}}_{\mathcal{M}}\mathbf{H}^{l-1}_{\mathcal{M}} \tag{10}$$

---

[3]For succinctness, we omit the layer and head enumeration

As in Vaswani et al. (2017), $\mathbf{H}'_{\mathcal{M}}$ and $\mathbf{H}'_{\mathcal{V}}$ are then augmented through skip connections and layer normalization, before being passed to the self-attention layer, yielding the agent and task embeddings $\mathbf{H}^l_{\mathcal{M}}$ and $\mathbf{H}^l_{\mathcal{V}}$, respectively, of the current layer $l$.

Given the final agent and task embeddings $\mathbf{H}_{\mathcal{M}}$ and $\mathbf{H}_{\mathcal{V}}$, respectively, the policy utilized in MAC-SIM uses a multiple-pointer mechanism similar to Berto et al. (2024):

$$\mathbf{L} = c \cdot \tanh\left(\frac{\mathbf{Q}\mathbf{K}^{\top}}{\sqrt{d}}\right), \quad \mathbf{Q} = \mathbf{W}^{\mathbf{Q}}_{\text{Dec}}\mathbf{H}_{\mathcal{M}}, \quad \mathbf{K} = \mathbf{W}^{\mathbf{K}}_{\text{Dec}}(\mathbf{H}_{\mathcal{V}}||\mathbf{h}^{\text{skip}}) \tag{11}$$

with learnable parameters $W^Q_{\text{Dec}}$ and $W^K_{\text{Dec}} \in \mathbb{R}^{d \times d}$, and $c$ is a scale parameter, set to 10 following Bello et al. (2017) to enhance exploration.

## C    PROBLEM DEFINITIONS

### C.1    FLEXIBLE JOB SHOP SCHEDULING PROBLEM

The Flexible Job Shop Scheduling Problem (FJSP) is a highly complex, NP-hard optimization problem and a generalization of the classical Job Shop Scheduling Problem (JSP). The problem concerns scheduling a set of $N$ jobs on $M$ machines. Each job consists of a sequence of operations that must be processed in a specific order. The critical distinction from the classical JSP is the "flexibility": each operation can be processed by any machine from a given subset of capable machines. This introduces two interdependent decision layers: a routing problem (assigning each operation to a suitable machine) and a sequencing problem (determining the order of operations on each machine). The FJSP can be modeled as a multi-agent CO problem, where each machine acts as an agent responsible for building its own schedule. In an autoregressive framework, a decision is made at each step to assign and schedule the next operation, considering machine availability and job precedence constraints. The ultimate goal is typically to find a schedule that minimizes the makespan, i.e., the time required to complete all operations for all jobs.

#### C.1.1    MATHEMATICAL MODEL

We present a common mixed-integer linear programming (MILP) model for the FJSP, based on the formulation described by Özgüven et al. (2010):

**Indices**

| | |
|---|---|
| $j, l$ | Job index |
| $i, h$ | Operation index |
| $k$ | Machine index |

**Parameters**

| | |
|---|---|
| $N$ | Number of jobs |
| $M$ | Number of machines |
| $O_j$ | Set of operations for job $j$ |
| $M_{ij}$ | Set of machines capable of processing operation $i$ of job $j$ |
| $p_{ijk}$ | Processing time of operation $i$ of job $j$ on machine $k$ |
| $B$ | A very large positive number (for big-M constraints) |

**Decision variables**

| | |
|---|---|
| $C_{max}$ | The makespan (maximum completion time) |
| $c_{ij}$ | Completion time of operation $i$ of job $j$ |
| $x_{ijk}$ | $\begin{cases} 1 & \text{if operation } i \text{ of job } j \text{ is assigned to machine } k \\ 0 & \text{otherwise} \end{cases}$ |
| $y_{jilkh}$ | $\begin{cases} 1 & \text{if op } (j, i) \text{ precedes op } (l, h) \text{ on machine } k \\ 0 & \text{otherwise} \end{cases}$ |

**Objective:**

$$\min C_{max} \tag{12}$$

**Subject to:**

$$C_{max} \geq c_{ij} \qquad\qquad \forall j,\ \forall i \in O_j \tag{13}$$

$$\sum_{k \in M_{ij}} x_{ijk} = 1 \qquad\qquad \forall j,\ \forall i \in O_j \tag{14}$$

$$c_{ij} - \sum_{k \in M_{ij}} p_{ijk} x_{ijk} \geq c_{(i-1)j} \qquad\qquad \forall j,\ \forall i \in O_j, i > 1 \tag{15}$$

$$c_{ij} - c_{lh} + B(1 - y_{jilkh}) \geq p_{ijk} - B(2 - x_{ijk} - x_{lhk}) \qquad \forall j, l, i, h, k \text{ s.t. } j < l \tag{16}$$

$$c_{lh} - c_{ij} + B \cdot y_{jilkh} \geq p_{lhk} - B(2 - x_{ijk} - x_{lhk}) \qquad \forall j, l, i, h, k \text{ s.t. } j < l \tag{17}$$

$$c_{ij} \geq \sum_{k \in M_{ij}} p_{ijk} x_{ijk} \qquad\qquad \forall j,\ \forall i \in O_j, i = 1 \tag{18}$$

The objective function in equation 12 minimizes the makespan. The makespan itself is defined by constraint set equation 13 as being greater than or equal to the completion time of every operation. Constraint set equation 14 is the assignment constraint, ensuring that each operation is assigned to exactly one machine from its set of eligible machines. The chronological sequence of operations within the same job is enforced by constraint set equation 15, which states that an operation $(j, i)$ cannot be completed before the completion of its predecessor $(j, i-1)$ plus its own processing time. For the first operation of a job, constraint set equation 18 ensures its completion time is at least its processing time. Constraint sets equation 16 and equation 17 are the core disjunctive constraints that prevent a machine from processing more than one operation at a time. For any two operations $(j, i)$ and $(l, h)$ assigned to the same machine $k$, these "big-M" constraints ensure that one must finish before the other begins. The binary variable $y_{jilkh}$ determines their relative order.

### C.1.2 MULTI-AGENT MDP

To solve the FJSP with MACSIM, we cast it as a multi-agent Markov decision process (MDP) as defined in Section 3. In this formulation, we define the **state** of the FJSP at construction step $t$ as a bipartite graph, where machines correspond to the set of agents $\mathcal{M}$ and jobs/operations to the set of tasks $\mathcal{V}$. An edge $(m, v) \in \mathcal{E}$ exists if machine $m$ can process the next operation of job $v$, with processing time $w(m, v)$. At each step, each agent $m$ chooses an **action** $a^m = v$ from the set of admissible edges $(m, v)$ (or the skip token), which assigns job $v$'s next operation to machine $m$. The joint action $\mathbf{a}_t = \{a_t^m\}_{m \in \mathcal{M}}$ induces a deterministic **transition** to $s_{t+1}$, which updates the schedule, the job progress, and machine queues. The **reward** is 0 for all intermediate steps, with a final terminal reward of the negative makespan ($-C_{max}$) shared between all agents.

### C.2 FLEXIBLE FLOW SHOP PROBLEM

The flexible flow shop problem (FFSP) is a challenging and extensively studied optimization problem in production scheduling, involving $N$ jobs that must be processed by a total of $M$ machines divided into $i = 1 \ldots S$ stages, each with multiple machines ($M_i > 1$). Jobs follow a specified sequence through these stages, but within each stage, any available machine can process the job, with the key constraint that no machine can handle more than one job simultaneously. The FFSP can naturally be viewed as a multi-agent CO problem by considering each machine as an agent that constructs its own schedule. Adhering to autoregressive CO, agents construct the schedule sequentially, selecting one job (or no job) at a time. The job selected by a machine (agent) at a specific stage in the decoding process is scheduled at the earliest possible time, that is, the maximum of the time the job becomes available in the respective stage (i.e., the time the job finished on prior stages) and the machine becoming idle. The process repeats until all jobs for each stage have been scheduled, and the ultimate goal is to minimize the makespan, i.e., the total time required to complete all jobs.

### C.2.1 MATHEMATICAL MODEL

We use the model outlined in Kwon et al. (2021) to define the FFSP:

**Indices**

$i$      Stage index
$j, l$      Job index
$k$      Machine index in each stage

**Parameters**

$N$      Number of jobs
$S$      Number of stages
$M_i$      Number of machines in stage $i$
$B$      A very large number
$p_{ijk}$      Processing time of job $j$ in stage $i$ on machine $k$

**Decision variables**

$c_{ij}$      Completion time of job $j$ in stage $i$

$x_{ijk}$      $\begin{cases} 1 & \text{if job } j \text{ is assigned to machine } k \text{ in stage } i \\ 0 & \text{otherwise} \end{cases}$

$y_{ilj}$      $\begin{cases} 1 & \text{if job } l \text{ is processed earlier than job } j \text{ in stage } i \\ 0 & \text{otherwise} \end{cases}$

**Objective:**

$$\min \left( \max_{j=1..n} \{c_{Sj}\} \right) \tag{19}$$

**Subject to:**

$$\sum_{k=1}^{M_i} x_{ijk} = 1 \qquad\qquad i = 1, \ldots, S; \; j = 1, \ldots, N \tag{20}$$

$$y_{iij} = 0 \qquad\qquad i = 1, \ldots, S; \; j = 1, \ldots, N \tag{21}$$

$$\sum_{j=1}^{N} \sum_{l=1}^{N} y_{ilj} = \sum_{k=1}^{M_i} \max \left( \sum_{j=1}^{n} (x_{ijk}) - 1, 0 \right) \qquad i = 1, \ldots, S \tag{22}$$

$$y_{ilj} \leq \max \left( \max_{k=1 \ldots M_i} \{x_{ijk} + x_{ilk}\} - 1, 0 \right) \quad i = 1, \ldots, S; \; j, l = 1, \ldots, N \tag{23}$$

$$\sum_{l=1}^{N} y_{ilj} \leq 1 \qquad\qquad i = 1, \ldots, S; \; j = 1, \ldots, N \tag{24}$$

$$\sum_{j=1}^{N} y_{ilj} \leq 1 \qquad\qquad i = 1, \ldots, S; \; l = 1, \ldots, N \tag{25}$$

$$c_{1j} \geq \sum_{k=1}^{m_1} p_{1jk} \cdot x_{1jk} \qquad\qquad j = 1, \ldots, N \tag{26}$$

$$c_{ij} \geq c_{i-1j} + \sum_{k=1}^{M_i} p_{ijk} \cdot x_{ijk} \qquad\qquad i = 2, 3, \ldots, S; \; j = 1, \ldots, N \tag{27}$$

$$c_{ij} + B(1 - y_{ilj}) \geq c_{il} + \sum_{k=1}^{M_i} p_{ijk} \cdot x_{ijk} \qquad\qquad i = 1, \ldots, S; \; j, l = 1, \ldots, N \tag{28}$$

Here, the objective function equation 19 minimizes the makespan of the resulting schedule, that is, the completion time of the job that finishes last. The schedule has to adhere to several constraints: First, constraint set equation 20 ensures that each job is assigned to exactly one machine at each stage. Constraint sets equation 21 through equation 25 define the precedence relationships between jobs within a stage. Specifically, constraint set equation 21 ensures that a job has no precedence relationship with itself. Constraint set equation 22 ensures that the total number of precedence relationships in a stage equals $N - M_i$ minus the number of machines with no jobs assigned. Constraint set equation 23 dictates that precedence relationships can only exist among jobs assigned to the same machine. Additionally, constraint sets equation 24 and equation 25 restrict a job to having at most one preceding job and one following job.

Moving on, constraint set equation 26 specifies that the completion time of a job in the first stage must be at least as long as its processing time in that stage. The relationship between the completion times of a job in consecutive stages is described by constraint set equation 27. Finally, constraint set equation 28 ensures that no more than one job can be processed on the same machine at the same time.

### C.2.2 MULTI-AGENT MDP

We formulate the FFSP as a multi-agent MDP. At construction step $t$, the **state** is represented by a bipartite graph: jobs form the task set $\mathcal{V}$, and machines grouped by production stages form the agent set $\mathcal{M}$. Within each stage, an edge $(m, v) \in \mathcal{E}$ exists if machine $m$ at that stage is eligible to process the next pending operation of job $v$, with processing time $w(m, v)$. At each step, each agent $m \in \mathcal{M}$ selects an **action** $a^m = v$ from its admissible edges $(m, v)$, assigning the corresponding job operation to that machine. The joint action $\mathbf{a}_t$ induces a deterministic **transition** to $s_{t+1}$ by updating the partial schedule, machine queues, and job progression across stages. Intermediate steps yield zero **reward**, while the terminal state provides a reward equal to the negative makespan shared between all agents.

### C.3 MIN-MAX HETEROGENEOUS CAPACITATED VEHICLE ROUTING PROBLEM

The min-max heterogeneous capacitated vehicle routing problem (HCVRP) is an NP-hard combinatorial optimization problem, representing a significant extension of the classic Capacitated Vehicle Routing Problem (CVRP). The problem involves designing a set of optimal routes for a heterogeneous fleet of vehicles, stationed at a central depot, to serve a geographically dispersed set of customers, each with a specific demand. The heterogeneity implies that vehicles may differ in their capacities and costs. The primary objective of the HCVRP is to minimize the length (or cost, or duration) of the longest single route in the solution, rather than the total length of all routes. This "Min-Max" criterion is crucial for applications where the balance of workload between drivers is a priority.

### C.3.1 MATHEMATICAL MODEL

We present a three-index vehicle flow formulation for the HCVRP, adapted from standard VRP models as described in works such as Li et al. (2022):

**Indices**

| | |
|---|---|
| $i, j$ | Node indices (customers and depot) |
| $k$ | Vehicle index |

**Sets**

| | |
|---|---|
| $L$ | Set of $N$ customers, indexed $1, \ldots, N$ |
| $V$ | Set of all nodes, $V = L \cup \{0\}$, where 0 is the depot |
| $K$ | Set of vehicles |

**Parameters**

| | |
|---|---|
| $d_i$ | Demand of customer $i \in L$ |
| $Q_k$ | Capacity of vehicle $k \in K$ |
| $c_{ijk}$ | Cost (e.g. travel time) for vehicle $k$ to travel between nodes $i$ and $j$ |

**Decision variables**

$C_{max}$      The maximum route length (the objective)

$x_{ijk}$      $\begin{cases} 1 & \text{if vehicle } k \text{ travels directly from node } i \text{ to node } j \\ 0 & \text{otherwise} \end{cases}$

$u_{ik}$      Continuous variable representing the load of vehicle $k$ after visiting node $i$

**Objective:**

$$\min C_{max} \tag{29}$$

**Subject to:**

$$C_{max} \geq \sum_{i \in V} \sum_{j \in V, i \neq j} c_{ijk} x_{ijk} \qquad \forall k \in K \tag{30}$$

$$\sum_{k \in K} \sum_{i \in V, i \neq j} x_{ijk} = 1 \qquad \forall j \in L \tag{31}$$

$$\sum_{i \in V, i \neq h} x_{ihk} = \sum_{j \in V, j \neq h} x_{hjk} \qquad \forall h \in L,\ \forall k \in K \tag{32}$$

$$\sum_{j \in L} x_{0jk} \leq 1 \qquad \forall k \in K \tag{33}$$

$$u_{ik} - u_{jk} + Q_k x_{ijk} \leq Q_k - d_j \qquad \forall i, j \in L, i \neq j,\ \forall k \in K \tag{34}$$

$$d_i \sum_{j \in V} x_{jik} \leq u_{ik} \leq Q_k \sum_{j \in V} x_{jik} \qquad \forall i \in L,\ \forall k \in K \tag{35}$$

The objective function eq. (29) minimizes the maximum route cost $C_{max}$. $C_{max}$ is defined by constraint set equation 30, which ensures it is greater than or equal to the calculated cost of every individual route, using the vehicle-specific cost parameter $c_{ijk}$.

Constraint set equation 31 guarantees that each customer is visited exactly once by one vehicle. The vehicle flow conservation is handled by two sets of constraints. First, constraint set equation 32 ensures that if a vehicle enters a customer node, it must also depart from it. Second, constraint set equation 33 ensures that each vehicle can leave the depot at most once; combined with constraint set equation 32, this also implies that any vehicle that serves customers must return to the depot.

Finally, constraint sets equation 34 and equation 35 are the Miller-Tucker-Zemlin constraints, which simultaneously prevent subtours and enforce vehicle capacity limits. The continuous variable $u_{ik}$ tracks the cumulative load of vehicle $k$. Constraint set equation 34 establishes a valid sequence for load accumulation, while constraint set equation 35 binds the load variable for each customer visit, ensuring it is positive only if the customer is on vehicle $k$'s route and that it never exceeds the vehicle's capacity $Q_k$.

### C.3.2   MULTI-AGENT MDP

We formulate the HCVRP as a multi-agent MDP where each vehicle is modeled as an agent. The **state** at construction step $t$ consists of a graph with customer nodes $\mathcal{V}$ and vehicles $\mathcal{M}$, where each customer $v \in \mathcal{V}$ has a remaining demand $d_{t,v}$, and each vehicle $m \in \mathcal{M}$ has residual capacity $c_{t,m}$ and a current position on the graph. At each step, each vehicle $m$ selects an **action** $a^m = v$ corresponding to the next customer to visit, a return to the depot, or a stay at the current location, subject to feasibility given its residual capacity and route status. The joint action $\mathbf{a}_t$ determines the **transition** to $s_{t+1}$ by updating vehicle positions, capacities, and customer demands. The shared **reward** is sparse: intermediate steps yield zero reward, while the final reward is defined as the negative of the maximum travel cost among all agents $-C_{max}$.

Table 5: Generalization performance on public FJSP benchmark instances.

| | FJSP | | | | | | | | | | | |
| | Brandimarte | | | Hurink (rdata) | | | Hurink (edata) | | | Hurink (vdata) | | |
| Metric | Obj. | Gap | Time (s.) | Obj. | Gap | Time (s.) | Obj. | Gap | Time (s.) | Obj. | Gap | Time (s.) |
|---|---|---|---|---|---|---|---|---|---|---|---|---|
| OR-Tools | **174.20** | 0.99% | 1447 | **935.80** | 0.16% | 1397 | **1028.93** | 0.00% | 899 | **919.60** | 0.00% | 639 |
| MWKR | 201.70 | 16.52% | 0.49 | 1053.10 | 12.72% | 0.52 | 1219.01 | 18.47% | 0.52 | 952.01 | 3.52% | 0.52 |
| PPO (g.) | 198.50 | 15.07% | 1.25 | 1030.83 | 10.33% | 1.4 | 1182.08 | 14.88% | 1.4 | 954.33 | 3.78% | 1.37 |
| DANIEL (g.) | 184.40 | 6.90% | 1.3 | 1031.63 | 10.42% | 1.37 | 1175.53 | 14.25% | 1.37 | 944.85 | 2.75% | 1.36 |
| SLIM (g.) | 195.12 | 13.11% | 2.16 | 1011.15 | 8.23% | 2.28 | 1170.45 | 13.75% | 2.29 | 937.04 | 1.90% | 2.28 |
| MACSIM (g.) | 185.80 | 7.71% | 1.12 | 992.10 | 6.19% | 0.87 | 1168.97 | 13.61% | 0.85 | 937.07 | 1.90% | 1.04 |
| PPO (s.) | 190.30 | 10.32% | 4.13 | 985.33 | 5.46% | 4.81 | 1116.68 | 8.53% | 4.87 | 930.80 | 1.22% | 4.72 |
| DANIEL (s.) | 180.80 | 4.81% | 4.12 | 978.28 | 4.71% | 4.73 | 1119.73 | 8.82% | 4.73 | 925.40 | 0.63% | 4.77 |
| SLIM (s.) | 191.20 | 10.84% | 30.75 | 963.55 | 3.13% | 30.81 | 1117.50 | 8.61% | 30.82 | 924.07 | 0.49% | 33.12 |
| MACSIM (s.) | 177.30 | 2.78% | 9.78 | 957.92 | 2.53% | 6.84 | 1094.85 | 6.41% | 7.05 | 923.17 | 0.39% | 8.83 |

## D  MORE EXPERIMENTS

### D.1  PUBLIC BENCHMARK DATASET FOR FJSP

In addition to the synthetic FJSP instances used in Tables 1 and 2, we analyze the generalization capabilities of MACSIM on two well-known FJSP benchmarks. First, the Brandimarte benchmark comprises 10 benchmark cases ranging from 10–20 jobs with 3–15 operations each, processed on 4–15 machines with varying machine flexibility (Brandimarte, 1993). Second, the benchmark from Hurink et al. (1994) contains 120 instances ranging from 6–30 jobs and 6–15 machines, with the number of operations per job matching the machine count. These are categorized into three groups based on machine flexibility:

- edata: Few operations may be assigned to more than one machine.
- rdata: Most of the operations may be assigned to some machines.
- vdata: All operations may be assigned to several machines.

The aggregated results for these 4 benchmarks are presented in Table 5. For each instance type, we report results for the best-performing model selected from the models trained on the instance sizes reported in Table 1. MACSIM consistently outperforms all other neural baselines and substantially narrows the gap to the state-of-the-art CP-SAT solver, OR-Tools.

### D.2  INCREASING THE NUMBER OF AGENTS IN FFSP

We further evaluate all solvers for the FFSP on instances with an increasing number of agents while keeping the number of jobs fixed. Specifically, we consider instances with $N = 50$ jobs processed across $S = 3$ stages, each containing $M_i = 4$, 6, and 8 machines, for a total of $M = 12$, 18, and 24 machines, respectively. The results in Table 6 reveal a widening performance gap between standard self-improvement (SLIM) and MACSIM, highlighting the importance of exploiting agent symmetries during training. These symmetries become more pronounced in problems with more agents, making coordination increasingly challenging. MACSIM consistently produces policies that achieve superior coordination as the number of agents grows.

### D.3  ANALYSIS OF THE SKIP TOKEN

In Figure 5a, we evaluate the effect of the skip token and its associated penalty on solution quality. The skip token proves essential for achieving high-quality solutions. However, introducing a penalty on its usage initially skews training towards solutions with fewer skip tokens, which results in slightly worse performance compared to MACSIM without the penalty. By annealing the penalty towards zero via an exponential decay, the model ultimately reaches solutions of equal quality while requiring significantly fewer construction steps (forward passes through $\pi_\theta$), as confirmed in Figure 5b. The different penalty schedules reported in Figure 5b are visualized in Figure 5c.

Although we found that the policy is largely insensitive to the specific form of penalty decay, we adopt an exponential decay because it naturally prevents the penalty from reaching exactly zero. If the penalty were to reach zero, the model would immediately resume increasing its use of the skip

Table 6: Test set performance on FFSP instances with varying number of agents.

| | FFSP | | | | | | | | |
|---|---|---|---|---|---|---|---|---|---|
| $N \times M_i \times S$ | $50 \times 4 \times 3$ | | | $50 \times 6 \times 3$ | | | $50 \times 8 \times 3$ | | |
| Metric | Obj. | Gap | Time (s.) | Obj. | Gap | Time (s.) | Obj. | Gap | Time (s.) |
| Shortest Job First | 56.94 | 19.27% | 0.37 | 38.01 | 23.65% | 0.25 | 29.39 | 27.45% | 0.25 |
| Genetic Algorithm | 56.92 | 19.23% | 44.33 | 38.26 | 24.46% | 47.12 | 29.05 | 25.98% | 50.95 |
| Particle Swarm Opt. | 55.1 | 15.42% | 82.74 | 36.83 | 19.81% | 85.43 | 28.06 | 21.68% | 89.01 |
| MatNet ($g.$) | 51.52 | 7.92% | 2.17 | 34.82 | 13.27% | 2.42 | 27.52 | 19.34% | 2.65 |
| PARCO ($g.$) | 51.19 | 7.23% | 0.52 | 32.88 | 6.96% | 0.50 | 24.89 | 7.94% | 0.44 |
| SLIM ($g.$) | 50.01 | 4.75% | 2.28 | 32.99 | 7.32% | 3.01 | 25.04 | 8.59% | 3.87 |
| MACSIM ($g.$) | 49.36 | 3.39% | 0.43 | 32.23 | 4.85% | 0.59 | 24.45 | 6.03% | 0.61 |
| MatNet ($s.$) | 49.68 | 4.06% | 8.91 | 33.45 | 8.82% | 9.23 | 26.00 | 12.75% | 9.81 |
| PARCO ($s.$) | 49.27 | 3.20% | 1.97 | 31.60 | 2.80% | 1.89 | 23.59 | 2.30% | 1.68 |
| SLIM ($s.$) | 48.13 | 0.82% | 8.87 | 31.65 | 2.96% | 9.64 | 24.41 | 5.85% | 10.6 |
| MACSIM ($s.$) | **47.74** | 0.00% | 0.91 | **30.74** | 0.00% | 1.65 | **23.06** | 0.00% | 1.78 |

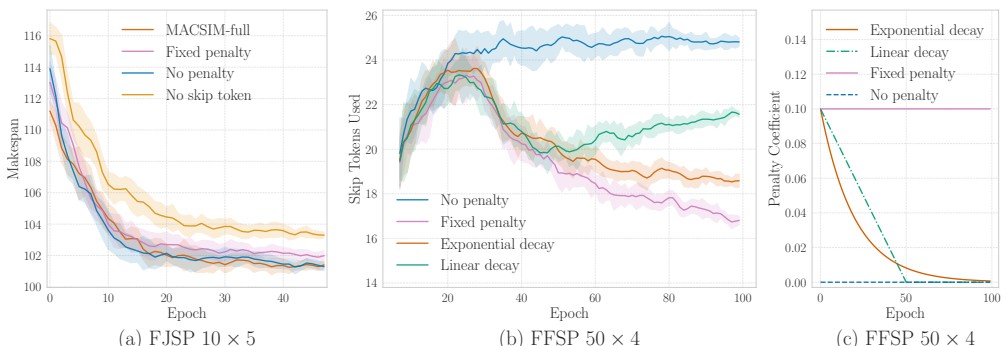

(a) FJSP $10 \times 5$     (b) FFSP $50 \times 4$     (c) FFSP $50 \times 4$

Figure 5: **Left**: Training curves for MACSIM with and without skip token as well as different penalties. **Middle**: Effect of different penalty strategies on the number of skip tokens used. **Right**: Skip token penalty coefficient for different strategies over the course of training.

token as shown in Figure 5b. Maintaining a penalty slightly above zero ensures that, when multiple solutions have equal objective values during training data generation, the skip token usage serves as a meaningful tie-breaking criterion for the expert data selection. As such, a very small penalty still serves as an effective regularizer during the later stages of training.

Moreover, Figure 5b illustrates how the model learns to regulate skip token usage. At the start of training, the model employs the skip token at a relatively low rate. It then rapidly increases its usage, recognizing that deferring a decision in FFSP can unlock better scheduling opportunities for machines. As training progresses, the model reduces unnecessary skips to avoid the penalty, returning to nearly the same skip frequency as at the beginning. Crucially, however, the model has now learned to distinguish when deferring an assignment is beneficial and when it is not.

### D.4 TRAINING DYNAMICS ANALYSIS

Figure 6 offers additional insight into the training behavior of MACSIM relative to SLIM across FFSP instances of increasing size. Beyond final performance metrics, these curves highlight fundamental differences in stability and efficiency. As the problem size grows from $N = 20$ to $N = 100$ jobs, SLIM exhibits increasingly unstable training dynamics, with pronounced oscillations in validation performance. In contrast, MACSIM maintains smooth and monotonic convergence. This contrast suggests that SLIM's single-action supervision becomes insufficient for complex coordination tasks, while MACSIM's set-based loss provides more reliable and stable gradient signals.

The comparison of wall-clock training times (right y-axis) further demonstrates that MACSIM's efficiency advantages extend to the training process itself. MACSIM trains nearly an order of magnitude faster per epoch while still achieving superior final performance. This consistent efficiency gain across problem sizes, coupled with its robustness on larger instances where SLIM becomes unstable, provides strong empirical evidence that the joint-action framework effectively overcomes scalability limitations in self-improvement for multi-agent combinatorial optimization.

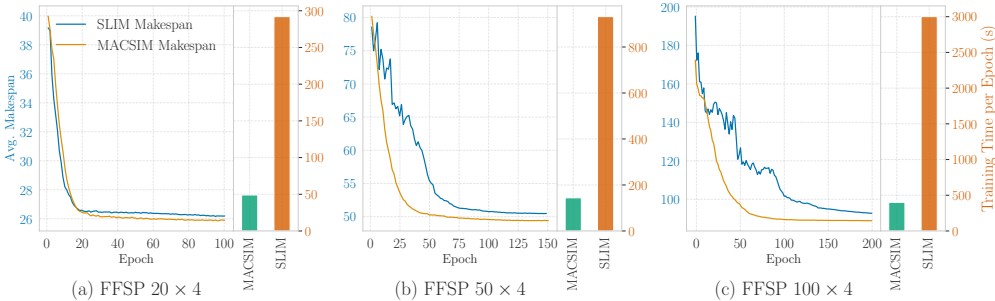

Figure 6: Wall-clock training time per epoch (right axis) and evolution of the average makespan on the validation set of FFSP instances with $N = 20, 50$, and $100$ jobs (left axis). As the problem size increases, SLIM's training becomes progressively unstable, whereas MACSIM maintains stability and converges while being nearly an order of magnitude faster to train.

## D.5 SCALABILITY ANALYSIS

A key concern in neural combinatorial optimization is scalability to large problem instances. In this section, we provide a comprehensive analysis demonstrating that MACSIM's design choices – training on single state-action pairs and amortizing re-encoding costs over multiple actions – yield substantial scalability advantages over both traditional RL-based NCO methods and standard self-improvement approaches.

**Comparison with RL-based NCO.** Traditional RL-based NCO methods like MatNet (Kwon et al., 2021) train via REINFORCE, which requires backpropagation through entire solution trajectories. This necessitates storing activations for all $T$ decoding steps, leading to memory consumption that scales linearly with solution length. In contrast, MACSIM's self-improvement paradigm trains on individual state-action pairs, requiring only single-step gradients.

We empirically validate this advantage by comparing memory consumption and training efficiency between MACSIM and MatNet on FFSP instances of increasing size. For memory analysis, we use a batch size of 1 for both methods and record peak GPU memory consumption during training. For training time analysis, we use the maximum feasible batch size for each method on a single NVIDIA A100 GPU with 40GB VRAM and measure wall-clock time per epoch (1,000 instances).

Results are presented in Table 7. For $N = 100$ jobs, MatNet consumes approximately 1 GB of memory, which increases to roughly 13 GB for $N = 500$, corresponding to a 13× increase. In contrast, MACSIM increases from only 702 MB to just under 2 GB – less than a 3× increase. This dramatic difference in memory scaling enables MACSIM to use significantly larger batch sizes (64 vs. 8 for $N = 500$), directly translating to faster training: MatNet's training time grows from 6.4 minutes per epoch at $N = 100$ to 264 minutes at $N = 500$, while MACSIM only increases from 5.1 minutes to 63 minutes.

These results demonstrate that MACSIM's state-action pair training paradigm provides a fundamental scalability advantage over trajectory-based RL methods, enabling efficient training on problem sizes where traditional approaches become prohibitively expensive.

Table 7: Memory consumption and training time comparison between MatNet (RL-based) and MACSIM on FFSP instances of increasing size. Fraction columns show MACSIM's cost relative to MatNet.

| | **FFSP** | | | | | |
|---|---|---|---|---|---|---|
| $N \times M_i \times S$ | Peak Memory (GB) | | | Training Time (min/epoch) | | |
| | MatNet | MACSIM | Fraction | MatNet | MACSIM | Fraction |
| $100 \times 4 \times 3$ | 1.47 | 0.71 | 0.48 | 6.43 | 5.12 | 0.79 |
| $250 \times 4 \times 3$ | 4.12 | 1.24 | 0.30 | 31.12 | 15.60 | 0.50 |
| $500 \times 4 \times 3$ | 12.87 | 1.95 | 0.15 | 264.41 | 63.62 | 0.24 |

**Comparison with Standard Self-Improvement.**  While standard self-improvement methods like SLIM share MACSIM's advantage of state-action pair training, they employ step-wise re-encoding with fully autoregressive decoding, thus computing new embeddings for every single action. MACSIM amortizes this computational cost by generating $M$ agent actions from a single encoding step.

Figure 3c in the main paper demonstrates this advantage across FFSP instances of varying size. For $N = 50$ jobs with $M = 24$ agents, MACSIM requires approximately 10 construction steps to generate a complete solution, while SLIM requires 150 steps. This translates directly to wall-clock time as shown in Figure 6: MACSIM trains in approximately 400 seconds per epoch compared to SLIM's 3,000 seconds – almost an order of magnitude faster.

**Combined Advantage: Scaling to large multi-agent Problems.**  The combination of (1) state-action pair training and (2) multi-action generation creates a powerful synergy for scaling to large multi-agent problems:

- Memory efficiency from state-action pair training enables large batch sizes during training

- Computational efficiency from amortized re-encoding significantly reduces training time by a factor proportional the number of agents

- Training stability from the permutation-invariant loss ensures reliable convergence even as coordination complexity increases

Together, these properties position MACSIM as a highly scalable approach for large-scale multi-agent combinatorial optimization.

### D.6    RUNTIME PROFILING AND EFFICIENCY ANALYSIS

To isolate the source of MACSIM's computational efficiency, we perform a decomposition of inference latency into two distinct components: (i) **policy evaluation**, which encompasses the forward pass of the neural network $\pi_\theta(\mathbf{L} \mid s_t)$ to generate logits, and (ii) **sampling**, which refers to the discrete selection of actions from the distribution $P(\mathbf{L})$.

We compare the baseline SLIM against MACSIM on the Flexible Flow Shop Problem (FFSP). We fix the problem size to $N = 50$ jobs across $S = 3$ stages and vary the number of parallel machines per stage, denoted as $M_i \in \{4, 6, 8\}$. This results in total agent counts of $M = 12, 18$, and $24$, respectively.

Table 8: **Inference time decomposition.** Times are reported as milliseconds for a single solution construction, averaged over 100 instances. The *Policy* column measures the cumulative time spent in neural network forward passes, while *Sampling* measures the time spent selecting discrete actions. Percentages indicate the relative contribution to total latency.

| | **FFSP** | | | |
| --- | --- | --- | --- | --- |
| $N \times M_i \times S$ | **MACSIM** | | **SLIM** | |
| | Policy | Sampling | Policy | Sampling |
| $50 \times 4 \times 3$ | 234.44 (65%) | 125.02 (35%) | 3119.36 (96%) | 120.12 (4%) |
| $50 \times 6 \times 3$ | 319.73 (71%) | 133.45 (29%) | 3966.12 (97%) | 120.40 (3%) |
| $50 \times 8 \times 3$ | 568.44 (79%) | 155.64 (21%) | 8844.75 (99%) | 121.01 (1%) |

**Policy.**  As shown in Table 8, inference in SLIM is overwhelmingly dominated by policy evaluation, which accounts for over 96% of total runtime. This bottleneck arises because SLIM operates as a single-action sequential predictor: for a problem requiring $N$ total assignments, SLIM must execute the encoder-decoder stack $N$ times.

In contrast, MACSIM reduces the policy evaluation time by a factor of 12 to 16. This speedup is a direct consequence of the multi-agent architecture defined in Equation (3), where the joint logit matrix $\mathbf{L}$ is computed once to construct $M$ actions. By enabling all $M$ agents to act based on a single forward pass, MACSIM effectively amortizes the expensive neural computation over $M$ decisions. Consequently, the end-to-end speedup ranges from approximately 9 to 11 across these settings.

**Sampling.** Notably, the absolute time spent on sampling remains comparable between the two methods (e.g., $\approx 125$ ms vs. $\approx 120$ ms for the $50 \times 3 \times 4$ instance), though MACSIM exhibits a slight increase as agent count grows. It is important to note that the total count of sampled actions in MACSIM is slightly higher than in SLIM, due to the incorporation of the *skip token*. While this mechanism increases the total volume of discrete samples required to complete a solution, the sampling cost remains negligible relative to the massive reduction in policy evaluation steps.

## E    LOSS FUNCTION GRADIENT ANALYSIS

Here, we provide a detailed analysis of the gradients for the four loss functions discussed and evaluated in the main text: the single-agent cross-entropy loss (evaluating only a single action $a_t^m \in \mathbf{a}_t$ at a time) akin to SLIM, the multi-agent cross-entropy loss ($\mathcal{L}_{\mathrm{CE}}$), the loss derived from the Plackett-Luce formulation ($\mathcal{L}_{\mathrm{PL}}$), and the loss corresponding to the Maximum Likelihood Estimation (MLE) of the generative model of Equation (3), denoted $\mathcal{L}_{\mathrm{ML}}$. This analysis highlights the sources of instability and bias in $\mathcal{L}_{\mathrm{SA}}$, $\mathcal{L}_{\mathrm{ML}}$, and $\mathcal{L}_{\mathrm{PL}}$, justifying our use of $\mathcal{L}_{\mathrm{CE}}$ as a permutation-invariant surrogate.

### E.1    SINGLE-AGENT CROSS-ENTROPY LOSS

This loss mimics standard self-improvement methods which are trained in a "next-token prediction" fashion as described by Pirnay & Grimm (2024). Given state $s_t$ and any pseudo-expert action $(m^*, v^*) \in \mathbf{a}_t^*$, the model's task under this loss is to predict this single pair from the entire, static pool of all possible agent-task assignments, $\mathcal{E}$. The loss is the negative log-likelihood of selecting the single expert pair $(m^*, v^*)$ from the set of all pairs $\mathcal{E}$:

$$\mathcal{L}_{\mathrm{SA}} = -\log \frac{\exp(L_{m^*,v^*})}{\sum_{(m,v)\in\mathcal{E}} \exp(L_{m,v})} \tag{36}$$

and the gradient for any logit $L_{i,j}$ in the logits matrix is given by:

$$\frac{\partial \mathcal{L}_{\mathrm{SA}}}{\partial L_{i,j}} = \frac{\exp(L_{i,j})}{\sum_{(m,v)\in\mathcal{E}} \exp(L_{m,v})} - \mathbb{I}(i = m^*, j = v^*) \tag{37}$$

where $\mathbb{I}(\cdot)$ is the indicator function.

**Conflicting and Spurious Gradients.** The core problem arises from the global softmax normalization. At each step, the gradient update for the target pair $(m_k^*, v_k^*)$ is defined in competition with *all other available pairs* in $\mathcal{E}$. This setup effectively designates a single pair as positive while treating the remaining $M - 1$ correct assignments as negatives. Consequently, the model is explicitly penalized for predicting other parts of the ground-truth solution: for every correct pair not chosen as the target, the gradient contributes a positive update that suppresses its logit. The resulting signal is inherently contradictory, preventing the model from converging toward a consistent joint policy.

This flaw is particularly evident in forced-choice scenarios. When an agent $m$ has only one valid task $v^*$ at step $k$, no genuine decision is required. Nevertheless, $\mathcal{L}_{\mathrm{ML}}$ still yields a non-zero loss, producing spurious gradients that penalize unrelated logits $L_{i,j}$ with $i \neq m$. Such unnecessary updates destabilize optimization by introducing noise unrelated to the actual decision process.

**Vanishing Gradients.** The softmax denominator is calculated over the entire set $\mathcal{E}$, which can contain thousands of possible actions. This makes the probability of the single target action infinitesimally small, leading to a vanishing gradient problem even more severe than that of the sequential MLE loss ($\mathcal{L}_{\mathrm{ML}}$). This effectively makes training infeasible for large problem sizes.

### E.2    MLE LOSS

Unlike the single-agent formulation, the MLE loss computes a softmax over the *remaining* feasible pairs at each step $k$, with $\mathcal{E}_k$ denoting the available pairs. The gradient for $L_{i,j}$ aggregates contributions across agents:

$$\frac{\partial \mathcal{L}_{\text{ML}}}{\partial L_{i,j}} = \sum_{k=1}^{M} \frac{\exp(L_{i,j})}{\sum_{(a,b) \in \mathcal{E}_k} \exp(L_{a,b})} - \mathbb{I}(i = m_k^*, j = v_k^*) \tag{38}$$

Similar to $\mathcal{L}_{\text{SA}}$, the global softmax normalization in $\mathcal{L}_{\text{ML}}$ leads to **conflicting and spurious gradient signals**. Since the loss is a sum over all agents, the final gradient for any correct assignment becomes an inefficient and contradictory sum of one large negative update and many small positive updates (penalties). Also, similar to $\mathcal{L}_{\text{SA}}$, this loss formulation suffers from **vanishing gradients**, especially for decisions made early in the sequence when the action space is at its largest. This effectively stalls the learning process, making it very difficult for the model to learn meaningful policies for large problems.

### E.3 PLACKETT-LUCE LOSS

The Plackett-Luce model addresses the global competition issue by conditioning the choice of a task on a specific agent at each step. If agent $i = m_k$ is assigned at step $k$, the softmax is computed only over the set of available tasks $\mathcal{V}_k = \mathcal{V} \setminus \{v_1, \ldots, v_{k-1}\}$. The gradient for agent $i$'s logit $L_{i,j}$ is:

$$\frac{\partial \mathcal{L}_{\text{PL}}}{\partial L_{i,j}} = \frac{\exp(L_{i,j})}{\sum_{j' \in \mathcal{V}_k} \exp(L_{i,j'})} - \mathbb{I}(j = v_k) \tag{39}$$

**Permutation-Induced Bias.** While $\mathcal{L}_{\text{PL}}$ resolves the conflicting gradient issue of $\mathcal{L}_{\text{ML}}$, it remains sensitive to the arbitrary order of the expert permutation. This bias arises directly from the structure of the gradient. For an agent acting early in a sequence, the set of available actions $\mathcal{V}_k$ is large, leading to a large normalization term in its softmax calculation. At the beginning of training, this large denominator yields a very small initial probability $P(v_k|i)$ for the expert action. Since a smaller probability results in a gradient with a larger magnitude, the optimization process is forced to drive the corresponding logit to a significantly higher magnitude. In contrast, a late-sequence agent starts with a higher probability and thus a smaller gradient magnitude, requiring a less substantial increase in its logit value. Consequently, the model dedicates capacity to learning an artificial, agent-specific confidence level, where the scale of the output logits becomes entangled with the agent's position in the training sequence.

### E.4 SET CROSS-ENTROPY LOSS

To avoid the issues of sequential modeling, the multi-agent cross-entropy loss ($\mathcal{L}_{\text{CE}}$) treats each agent's assignment as an independent classification problem. Each agent $i$ predicts its task $v^*(i)$ from the complete set of tasks $\mathcal{V}$. The gradient for agent $i$'s logit $L_{i,j}$ depends only on its own logits:

$$\frac{\partial \mathcal{L}_{\text{CE}}}{\partial L_{i,j}} = \frac{\exp(L_{i,j})}{\sum_{j' \in \mathcal{V}} \exp(L_{i,j'})} - \mathbb{I}(j = v^*(i)) \tag{40}$$

where $v^*(i)$ is the expert-assigned task for agent $i$.

This decoupled formulation provides a stable and robust learning signal. Every correct pair $(i, v_i^*)$ is reinforced individually, without interference or competition from the assignments of other agents. This elegant structure entirely circumvents the conflicting gradients of $\mathcal{L}_{\text{ML}}$ and the permutation bias of $\mathcal{L}_{\text{PL}}$, making it a superior objective for learning multi-agent assignment policies. Figure 6 validates the superiority of MACSIM trained with $\mathcal{L}_{\text{CE}}$ over SLIM that uses a single-agent loss $\mathcal{L}_{\text{SA}}$, especially for larger instances.

## F THEORETICAL JUSTIFICATION FOR SET-BASED LOSS IN MACSIM

### F.1 TRAINING-INFERENCE MISMATCH

The surrogate loss in equation 6 offers a tractable and stable method for training. While the generative model in equation 3 couples assignments through a one-to-one matching constraint, the set-based cross-entropy loss simplifies this by treating each expert agent-task pair as an independent

classification problem. This is justified because the training process only observes *valid matchings*, allowing for clean, per-agent supervision. The result is a permutation-invariant learning signal that provides stable gradients, avoiding the conflicts and biases inherent in more complex objectives like direct Maximum Likelihood Estimation (MLE) as shown in Appendix E.

This independent loss signal does not prevent the model from learning coordination. Inter-agent dependencies are captured implicitly by the policy's architecture, where self- and cross-attention mechanisms compute the joint logit matrix $\mathbf{L}$ with full context. At inference, the autoregressive sampling algorithm then acts as a hard constraint enforcer, translating these context-aware logits into a valid, conflict-free assignment. This combination of a stable surrogate for training and a structurally-aware process for inference proves highly effective. Empirically, this approach significantly outperforms training with an exact MLE or Plackett-Luce loss (Table 4), confirming its practical advantages.

In the following, we provide a formal justification for using the set-based cross-entropy loss, $\mathcal{L}_{\text{CE}}$. We prove that it is a sound proxy for the ideal (but intractable) permutation-invariant objective.

### F.2 Formal Justification for the Surrogate Loss

**Setup.** We introduce shorthand notation for softmax normalization and marginals. For agent $m$ and its expert-assigned task $v^*(m)$, we use:

$$Z_m = \sum_{v \in \mathcal{V}} \exp(L_{m,v}), \qquad p_m(v) = \frac{\exp(L_{m,v})}{Z_m}, \qquad p_m := p_m(v^*(m)).$$

During generation, the policy samples agents in some order $\sigma$ (a permutation of $\{1, \ldots, M\}$) with prior $w(\sigma)$, where $\sum_\sigma w(\sigma) = 1$. At step $k$, agent $m_k = \sigma(k)$ selects a task from the remaining set $\mathcal{V}_k(\sigma) \subseteq \mathcal{V}$. The normalizer for the softmax function is given as:

$$S_k(\sigma) = \sum_{v \in \mathcal{V}_k(\sigma)} \exp(L_{\sigma(k),v}).$$

The probability of producing the unordered matching $\mathbf{a}^*$ is

$$P_{\text{perm}}(\mathbf{a}^*) = \sum_\sigma w(\sigma) \prod_{k=1}^{M} \frac{\exp(L_{\sigma(k),v^*(\sigma(k))})}{S_k(\sigma)}.$$

The *ideal loss* is $\mathcal{L}_{\text{ideal}} = -\log P_{\text{perm}}(\mathbf{a}^*)$, while the surrogate loss is the set cross-entropy $\mathcal{L}_{\text{CE}} = -\sum_{m=1}^{M} \log p_m$.

**Theorem 2** (Upper Bound and Tightness). *For any permutation prior $w(\sigma)$:*

1. *(Upper Bound) The surrogate loss upper-bounds the ideal loss:*

$$\mathcal{L}_{CE} \geq \mathcal{L}_{\text{ideal}}.$$

2. *(Gap Bound) With $r_m = (1 - p_m)/p_m$:*

$$0 \leq \mathcal{L}_{CE} - \mathcal{L}_{\text{ideal}} \leq \sum_{m=1}^{M} \log(1 + r_m).$$

*If $p_m \geq 1 - \varepsilon$ for all m, then*

$$\mathcal{L}_{CE} - \mathcal{L}_{\text{ideal}} \leq M \log\left(1 + \tfrac{\varepsilon}{1-\varepsilon}\right) \leq \tfrac{M\varepsilon}{1-\varepsilon}.$$

**Part 1: Proof of the Upper Bound.**

*Proof.* Fix a permutation $\sigma$ and denote $P(\mathbf{a}^* \mid \sigma)$ its sequence probability. At each step $k$,

$$S_k(\sigma) = \sum_{v \in \mathcal{V}_k(\sigma)} \exp(L_{\sigma(k),v}) \leq \sum_{v \in \mathcal{V}} \exp(L_{\sigma(k),v}) = Z_{\sigma(k)},$$

since $\mathcal{V}_k(\sigma) \subseteq \mathcal{V}$. Thus

$$\frac{\exp(L_{\sigma(k),v^*(\sigma(k))})}{S_k(\sigma)} \geq \frac{\exp(L_{\sigma(k),v^*(\sigma(k))})}{Z_{\sigma(k)}} = p_{\sigma(k)}.$$

Hence

$$P(\mathbf{a}^* \mid \sigma) \geq \prod_{m=1}^{M} p_m.$$

Taking the expectation over $\sigma$ preserves this bound:

$$P_{\mathrm{perm}}(\mathbf{a}^*) = \sum_{\sigma} w(\sigma) P(\mathbf{a}^* \mid \sigma) \geq \prod_{m=1}^{M} p_m.$$

Applying $-\log(\cdot)$ yields

$$\mathcal{L}_{\mathrm{ideal}} \leq \mathcal{L}_{\mathrm{CE}}.$$

$\square$

### Part 2: Proof of the Gap Bound.

*Proof.* To bound the gap, we establish an upper bound on $P_{\mathrm{perm}}(\mathbf{a}^*)$. We start by rewriting the probability of a single sequence by multiplying and dividing each factor by $Z_{\sigma(k)}$:

$$P(\mathbf{a}^* \mid \sigma) = \prod_{k=1}^{M} \left( \frac{\exp(L_{\sigma(k),v^*(\sigma(k))})}{Z_{\sigma(k)}} \cdot \frac{Z_{\sigma(k)}}{S_k(\sigma)} \right) = \left( \prod_{k=1}^{M} p_{\sigma(k)} \right) \left( \prod_{k=1}^{M} \frac{Z_{\sigma(k)}}{S_k(\sigma)} \right).$$

Since $v^*(\sigma(k)) \in \mathcal{V}_k(\sigma)$, it must hold that $S_k(\sigma) \geq \exp(L_{\sigma(k),v^*(\sigma(k))})$, and thus:

$$\frac{Z_{\sigma(k)}}{S_k(\sigma)} \leq \frac{Z_{\sigma(k)}}{\exp(L_{\sigma(k),v^*(\sigma(k))})} = \frac{1}{p_{\sigma(k)}} = 1 + r_{\sigma(k)}.$$

Thus

$$P(\mathbf{a}^* \mid \sigma) \leq \left( \prod_{i=1}^{M} p_i \right) \left( \prod_{i=1}^{M} (1 + r_i) \right).$$

Averaging over $\sigma$ preserves the bound, so

$$P_{\mathrm{perm}}(\mathbf{a}^*) \leq \left( \prod_{m=1}^{M} p_m \right) \left( \prod_{m=1}^{M} (1 + r_m) \right).$$

Taking $-\log$ gives

$$\mathcal{L}_{\mathrm{ideal}} \geq \mathcal{L}_{\mathrm{CE}} - \sum_{m=1}^{M} \log(1 + r_m),$$

or equivalently

$$\mathcal{L}_{\mathrm{CE}} - \mathcal{L}_{\mathrm{ideal}} \leq \sum_{m=1}^{M} \log(1 + r_m).$$

If $p_m \geq 1 - \varepsilon$, then $r_m \leq \frac{\varepsilon}{1-\varepsilon}$, so

$$\sum_{m=1}^{M} \log(1 + r_m) \leq M \log\left(1 + \tfrac{\varepsilon}{1-\varepsilon}\right) \leq \tfrac{M\varepsilon}{1-\varepsilon}.$$

$\square$

**Implication for Optimization.** Theorem 2 establishes that $\mathcal{L}_{\mathrm{CE}}$ is a tractable upper bound on the true symmetric objective. Moreover, the bound tightens as the policy becomes confident, ensuring that optimization with $\mathcal{L}_{\mathrm{CE}}$ smoothly transitions from providing stable gradients early to closely approximating the ideal objective in later training.

## G  DISCUSSING THE INDEPENDENCE ASSUMPTION IN THE SET-BASED LOSS

This section analyzes the efficacy of the agent-independent loss function ($\mathcal{L}_{\mathrm{CE}}$) for solving coupled multi-agent combinatorial problems. The proposed loss function $\mathcal{L}_{\mathrm{CE}}$ treats the assignment of each agent $m$ to a task $v$ as an independent classification problem. Letting the total loss be the sum of individual agent losses, $\mathcal{L}_{\mathrm{CE}} = \sum_{m=1}^{M} \mathcal{L}_m(L_{m,:})$, a direct analysis of the gradients (Appendix E.4) shows that the loss for a specific agent $m$ yields no direct gradient signal with respect to the logits of any other agent:

$$\frac{\partial \mathcal{L}_m}{\partial L_{k,v}} = 0 \quad \forall k \neq m \tag{41}$$

Consequently, at the output layer, the optimization objective for Agent $m$ is independent of the decisions of Agent $k$. Despite this independence at the loss level, MACSIM achieves coordinated solutions through the synergy of three distinct mechanisms: **Architectural Coupling**, **Implicit Constraint Learning**, and **Inference-Time Constraints** enforced via our autoregressive sampling approach.

### G.1  MECHANISM 1: ARCHITECTURAL COUPLING

While the gradients are independent with respect to the logits $\mathbf{L}$, the agents are coupled via the shared model parameters $\theta$. The policy architecture processes the state through a deep encoder utilizing multiple layers of self-attention and cross-attention. In this shared latent space, the embedding of Agent $m$, denoted as $h_m$, is computed by aggregating information from all other agents and tasks. As a result, the gradient of Agent $m$'s loss with respect to the shared parameters affects the representations of all agents:

$$\frac{\partial \mathcal{L}_m}{\partial \theta} \neq 0 \tag{42}$$

This implies updates to features utilized by Agent $k$. To minimize $\mathcal{L}_{\mathrm{CE}}$, the encoder learns to produce disentangled representations where the features for Agent $m$ align with Task $v_m$, while the features for Agent $k$ align with Task $v_k$.

### G.2  MECHANISM 2: IMPLICIT CONSTRAINT LEARNING VIA GRADIENT DYNAMICS

The second coordination mechanism emerges from the specific gradient dynamics of the set-based loss. As detailed in our gradient analysis in Appendix E, single-action supervision (e.g., SLIM) inherently suffers from gradient interference. Because it utilizes a global Softmax over all possible agent-task pairs, reinforcing a single target action explicitly degrades the probability of valid future actions for other agents. This creates conflicting signals that hinder the learning of a consistent joint policy. In contrast, MACSIM avoids this interference by optimizing $M$ independent local losses. For any specific agent $m$, the objective maximizes the logit for the assigned task $v_m^*$ while suppressing the logits for all other tasks. Crucially, this set of suppressed tasks includes all tasks $v_k^*$ assigned to other agents $k \neq m$.

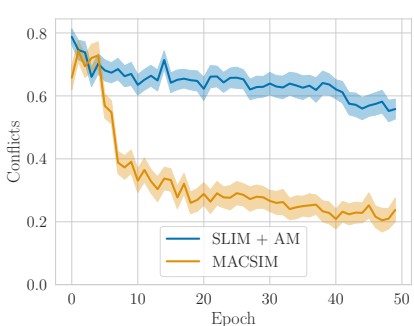

Figure 7: Evolution of the raw conflict rate (percentage of agents that share the same argmax action) during training on FJSP $10 \times 5$ instances.

Since the expert targets are conflict-free, *exactly one agent receives a positive update for any given task $v$, while all $M - 1$ other agents receive negative updates for that same task*, actively reducing contention. The policy internalizes global coordination constraints purely through this independent contrast.

We visualize this effect in Figure 8 and quantify it in Figure 7. Figure 8 compares the logit heatmaps of MACSIM against an Attention Model (AM) trained via standard SLIM. After training, MACSIM's logits exhibit a clear "permutation matrix" structure—high values are sparse and

non-overlapping across agents. Conversely, the AM baseline fails to coordinate, frequently assigning the highest logit value for a single task to multiple agents simultaneously.

Furthermore, we analyse the percentage of agents that share the same argmax action of the logits during training. As shown in Figure 7, this conflict rate drops precipitously during training for MACSIM while staying consistently on a high level for the AM trained via SLIM. This empirically validates that the independent loss, when combined with conflict-free expert labels, successfully teaches the network to respect injectivity constraints.

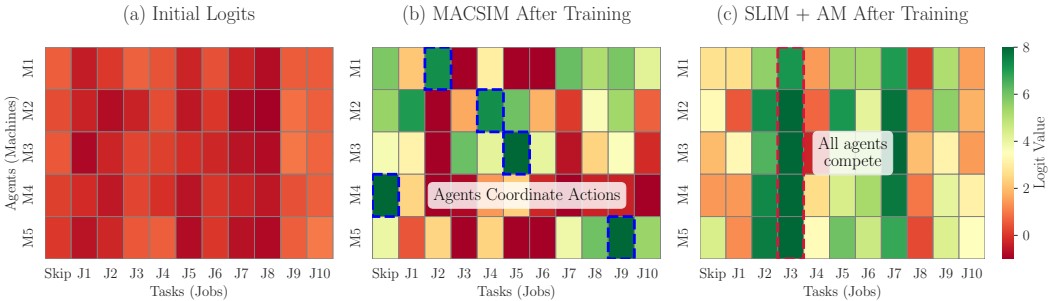

Figure 8: Comparison of joint logits $\mathbf{L}$ computed on an FJSP $10 \times 5$ problem instance by (a) an untrained MACSIM policy, (b) MACSIM after training, and (c) an Attention Model trained by SLIM.

### G.3 MECHANISM 3: INFERENCE-TIME CONSTRAINTS

The third layer of coordination occurs during the solution generation phase. Although the loss is calculated independently, the inference process defined in Algorithm 1 explicitly couples the agents via autoregressive sampling from the flattened joint logits. Because we sample without replacement, an agent that is highly confident (possessing a high logit value) exerts a "soft" suppression on all other agents. Specifically, if Agent $m$ has a high probability for Task $v$, it is statistically likely to be sampled early in the sequence. Once $(m, v)$ is added to the solution, task $v$ is masked. This mechanism effectively converts confidence into priority. An agent that was reinforced multiple times during training to take a specific action will exhibit a sharp probability peak. In our AR-sampling algorithm, this confident agent will take precedence, thereby preventing other agents from making a conflicting choice.

The ablation study in Table 3 validates that the policy implicitly learns to rank agents: replacing our logit-dependent sampling order with a fixed or random order significantly degrades performance, particularly for instances with a larger number of agents, proving that the learned confidence hierarchy is essential for resolving complex inter-agent conflicts.

## H EXPERIMENTAL DETAILS AND RESOURCES

### H.1 FJSP

#### H.1.1 HYPERPARAMETERS

We train MACSIM on the FJSP for 50 epochs using 4,000 randomly generated instances per epoch. For each training instance $x$ the current best policy $\pi_\theta^*$ is used to sample $\beta = 128$ solutions, where the best serves as training example. Within the training loop, we sample pseudo expert state-assignment pairs in batches of size 2,000 from the generated training dataset and use the Adam optimizer Kingma & Ba (2015) with a learning rate of $10^{-4}$, which we alter during training using a cosine annealing scheme. The embedding dimension $d$ is set to 256, transformer layers use 8 heads and we set the number of encoder layers $P$ to 4. Also, we use a dropout rate of 10%.

#### H.1.2 DATASETS

**Train data generation.** We train MACSIM on FJSP instances of size $10 \times 5$, $20 \times 5$, and $15 \times 10$, following the generation scheme and parameters described in Song et al. (2022).

**Testing.** Testing on the instance-types reported in Tables 1 and 2 is performed on 1000 separate test instances provided by Song et al. (2022). Further, we test MACSIM on public benchmark datasets (see Appendix D.1 for details).

## H.2 FFSP

### H.2.1 HYPERPARAMETERS

In each epoch, we train the models using 1,000 randomly generated instances for which we sample $\beta = 128$ solutions and put the best into the training dataset. During training, we sample pseudo expert state-assignment pairs in batches of size 1,000 from the generated training dataset and use the Adam optimizer Kingma & Ba (2015) with a learning rate of $10^{-4}$, which we alter during training using a cosine annealing scheme. Same as for the FJSP, the embedding dimension $d$ is set to 256 and we set the number of encoder layers $P$ to 4. Also, we use the same dropout rate of 10%. However, following Kwon et al. (2021) we use 16 heads in the attention layers. We train models corresponding to environments with 20 jobs for 100, with 50 jobs for 150 and with 100 jobs for 200 epochs.

### H.2.2 DATASETS

**Train data generation.** We follow the instance generation scheme outlined in Kwon et al. (2021) and sample processing times for job-machine pairs independently from a uniform distribution within the bounds $[2, 10]$. For the FFSP instance types shown in Table 1 we also use the same instance sizes as Kwon et al. (2021) with $N = 20, 50$ and 100 jobs and $M = 12$ machines which are spread evenly over $S = 3$ stages. To test for agent sensitivity in the FFSP, we fix the number of jobs to $N = 50$ but alter the number of agents for the last three instance types shown in Table 1. Still, we use $S = 3$ for this experiment, but alter the number of machines per stage to $M_i = 6, 8$ and 10, yielding a total of 18, 24 and 30 agents, respectively.

**Testing.** Testing on the instance-types reported in Table 1 is performed on 1000 separate test instances provided by Kwon et al. (2021). For the instance types reported in Table 6, test instances are generated randomly according to the above generation scheme.

## H.3 HCVRP

### H.3.1 HYPERPARAMETERS

We train MACSIM on the HCVRP for 200 epochs. In each epoch, we train the models using 1,000 randomly generated instances for which we sample $\beta = 200$ solutions. Within the training loop we sample pseudo expert state-assignment pairs in batches of size 1,000 from the generated training dataset and use Adam with a learning rate of $10^{-4}$, which we alter during training using a cosine annealing scheme. The embedding dimension $d$ is set to 256, dropout rate is 10%, transformer layers use 16 heads and we set the number of encoder layers $P$ to 4.

### H.3.2 DATASETS

**Train data generation.** We apply the instance generation scheme and seed used by Liu et al. (2024). The authors follow the standard procedure in NCO literature and sample coordinates for the $N$ customer locations and the depot from the unit square. The demand of customer locations is sampled i.i.d. from $\mathcal{U}(1, 10)$ and the capacity for each vehicle from $\mathcal{U}(20, 41)$. The speed of each vehicle is uniformly distributed within the range $\mathcal{U}(0.5, 1.0)$.

**Testing.** Testing is performed on the 1280 instances per $N \times M$ test setting from Liu et al. (2024). Neural baselines in Table 1 were trained with the specific number of nodes $N$ and number of agents $M$ they were tested on.

### H.4 Hardware and Software

#### H.4.1 Hardware

We experiment on a workstation equipped with 2 Intel(R) Xeon(R) Gold 6338 CPUs and 8 NVIDIA A100 graphic cards with 80 GB of VRAM each. Each training run uses a single A100.

#### H.4.2 Software

Our code base is implemented in Python 3.10. Neural policies are implemented in PyTorch 2.8 (Paszke et al., 2019) and training algorithms are defined as PyTorch Lightning Modules (Falcon & The PyTorch Lightning team, 2019). Environment implementations are based on or inspired by the RL4CO library (Berto et al., 2025). The operating system is Ubuntu 24.04 LTS.

### H.5 Use of Large Language Models

Large language models (LLMs) were used in this work solely as a general-purpose writing assistant. Their role was limited to polishing phrasing, improving clarity, and correcting grammar in drafts of the manuscript. All research ideas, analyses, results, and interpretations were generated and verified by the authors. The content produced by LLMs was carefully reviewed, edited, and integrated by the authors to ensure accuracy and adherence to academic standards.

