# OpenReview forum: "Multi-Action Self-Improvement For Neural Combinatorial Optimization"
_ICLR.cc/2026/Conference — ICLR 2026 Poster_

### Official Review · Reviewer_z3tB · 2025-10-28

**Soundness:** 3
**Presentation:** 3
**Contribution:** 3
**Rating:** 6
**Confidence:** 4

**Summary:**

This paper presents a novel approach to solving combinatorial optimization problems that require simultaneous assignment between multiple agents and multiple tasks over time. Such assignment problems can generally be formulated as bipartite matching problems.
The authors propose a method that jointly embeds multi-agent and multi-task information to generate joint logits, performs assignment based on these logits, and trains a neural network for joint logit embedding using a self-improvement learning framework.
Experiments are conducted on FJSP, FFSP, and HCVRP combinatorial optimization problems, showing faster inference and higher solution quality compared to previous studies.

**Strengths:**

- The strategy of assigning all agents and tasks simultaneously through a single joint logit output is efficient and demonstrates strong performance.

- The proposed approach is expected to be useful for solving other similar types of combinatorial optimization problems or for inspiring further research in this area.

- The training procedure for producing joint logits is logically well-structured.

- The paper is overall well-organized and highly readable.

**Weaknesses:**

- The overall model structure based on joint logits is similar to prior works (e.g., Kwon et al., 2021), and its novelty is mainly limited to the use of joint logits for agent–task assignment and the application of self-improvement for model training.

- The application of self-improvement does not introduce substantial new ideas or enhancements, and it appears to largely adopt existing approaches without major modifications.

**Questions:**

None

---

> ### Author Response · Authors · 2025-11-20
>
> Dear reviewer z3tB,
>
> we thank you for your time reviewing our paper and your positive evaluation of our work, particularly for acknowledging the efficiency and significance of our contribution. We address your concerns regarding novelty below.
>
> ### On the Novelty of Joint Logits and Multi-Agent Assignment
> ---
> While we build upon and combine established architectural components (like MatNet, and 2d-Ptr), the core innovation lies not in how we compute the joint logits, but in **how we exploit them** for multi-agent coordination and efficient action generation. Previous work either:
>
> - 2d-Ptr (Liu et al., 2024): Samples only once from joint distributions, discarding learned correlations [2].
> - PARCO (Berto et al., 2025): Assumes agent independence, leading to conflicts that require post-hoc resolution with breaking gradient signals.
>
> Our key contributions w.r.t. the multi-agent policy are:
>
> 1. **Autoregressive sampling without replacement (Algorithm 1)**: We prove this defines a valid probability distribution (Proposition 1) that captures inter-agent dependencies while guaranteeing conflict-free assignments. This is fundamentally different from prior approaches.
> 2. **Set-based learning objective**: We provide rigorous theoretical justification (Theorem 2, Section F) showing our surrogate loss is a tight upper bound on the intractable permutation-invariant objective, with the gap vanishing as the policy improves. This addresses a fundamental limitation in how multi-agent symmetries are exploited during learning.
> 3. **Enhanced Multi-step prediction via Skip-Token**: To make multi-step prediciton effective, we introduce a skip-token as a means for coordinated action deferral, which significantly increases the solution quality -- especially for problem with many agents where coordination is more difficult (Table 3). In Appendix D.3 we show how the policy learns to use the skip-token when using an annealed penalty factor.
>
> Unlike token-by-token generation, our policy amortizes computation by producing $M$ assignments per forward pass, achieving ~10× inference speedup (Figure 3c) while improving coordination.
>
> ### On Self-Improvement Contributions
> ---
> We respectfully disagree that our application of self-improvement (SI) lacks substantial innovation. Standard SI fundamentally breaks down for multi-agent problems due to agent-permutation symmetries (Figure 1). Consider a vehicle routing example: assigning driver 1→A, driver 2→B yields the identical solution as driver 2→B, driver 1→A, yet next-token prediction penalizes one order as "wrong."
> Our contributions to self-improvement are:
>
> - **First extension to multi-agent action spaces**: We reformulate the paradigm from $P(a_t | s_t)$ to $P(\[a^k_t\]_{k=1}^{M} | s_t)$, fundamentally changing what the policy learns.
>
> - **Permutation-invariant learning**: Our set-based loss (Section 4.4) eliminates the arbitrary agent ordering bias inherent in sequential supervision.
>
> - **Theoretical Validation** We provide extensive gradient analysis (Appendix E) proving this resolves conflicting signals in next-token loss $L_\text{SA}$ as well as in $L_\text{ML}$ and $L_\text{PL}$. Moreover, Appendix F shows how this loss bounds the ideal but intractable permutation invariant loss.
>
> - **Empirical validation**: These innovations translate to consistent improvements:
>
>    - Tables 1, 2, 5, and 6: MACSIM outperforms SLIM across all problems
>    - Figure 3b and Table 6: Performance gap increases with agent count (6% lower makespan than SLIM and 12%(!) lower than MatNet for 24 agents), validating our symmetry-aware approach
>    - Figure 6: MACSIM maintains stable training where SLIM becomes unstable on larger instances Table 2: Superior generalization to unseen problem sizes
>
> ### Positioning Our Work
> ---
> Our contribution is not merely applying existing techniques, but rather identifying and solving a fundamental limitation in how self-improvement handles multi-agent coordination. The combination of joint-action generation, conflict-free sampling, coordinated action deferral via the Skip-Token, and permutation-invariant learning creates a cohesive framework that advances the state-of-the-art for an important class of CO problems.
>
>
> We hope this clarifies the fundamental distinction to prior work and the motivation behind our design. We appreciate the opportunity to elaborate on this point and are happy to provide further explanation if needed.
>
> Sincerely,
> MACSIM authors

---

> > ### Comment · Reviewer_z3tB · 2025-11-27
> >
> > I appreciate the author's detailed and thoughtful rebuttal, which has addressed my concerns. However, my initial rating was already positive, and I believe it still reflects my overall assessment, so I will maintain the rating as is.

---

> > > ### Author Response · Authors · 2025-11-28
> > >
> > > Thank you for your time to go through our rebuttal. We are glad that we were able to address all your concerns, and we appreciate your positive assessment of our work as well as your constructive feedback.

---

### Official Review · Reviewer_tkUT · 2025-10-31

**Soundness:** 3
**Presentation:** 3
**Contribution:** 3
**Rating:** 6
**Confidence:** 3

**Summary:**

The paper proposes MACSIM, a method to solve multi-agent CO problems, inspired by recent self-improvement methods for neural CO. More concretely, MACSIM learns a multi-agent policy for joint agent-task assignments, where the policy is learned via self-improvement: the policy is trained to imitate previously found good solutions. Through experiments on the FJSP, FFSP, and HCVRP problems, the authors show the efficacy of their method against relevant baselines.

**Strengths:**

- The paper is mostly well-written.
- The problem studied in this paper (multi agent CO) is of interest to the neural CO community and the proposed method (self-improvement with a joint policy) is novel.
- The empirical results are good.

**Weaknesses:**

- The authors claim that prior self-improvement methods assume a unique optimal action per step, which limits their applicability to multi-agent settings with symmetric solutions. However, since the policy here is stochastic, it’s unclear to me why it couldn’t naturally capture multiple optimal actions if the imitation data already reflects these symmetries. Could the authors clarify this?

**Questions:**

See weaknesses.

---

> ### Author Response · Authors · 2025-11-20
>
> Dear reviewer tkUT,
>
> we thank you for your time reviewing our paper and your positive evaluation of our work, particularly regarding the novelty of our approach, the quality of empirical results, and the relevance to the neural CO community. We appreciate your question regarding why prior self-improvement (SI) methods with stochastic policies cannot naturally capture multiple optimal actions arising from agent-permutation symmetries. We clarify this below.
>
> ### Why Stochasticity in Standard SI Is Not Enough
> ---
> Although SI policies are stochastic at inference time, the training signal in standard SI fundamentally enforces a single optimal action per state.
>
> In existing SI methods (e.g., SLIM), training proceeds as follows:
>
> 1. Sample β solutions per instance
> 2. Select the **single best solution** $\tau^\ast$
> 3. Extract state-action pairs: $((s_1, a_1^\ast), (s_2, a_2^\ast), \ldots, (s_T, a_T^\ast))$
> 4. Train via cross-entropy loss: $-\text{log} \, P(a_t^\ast | s_t)$
>
> The critical limitation is that **only one action per state** enters the training dataset. Even though the policy is stochastic, it is explicitly supervised to predict this single action and penalized for predicting alternatives, including other actions that are part of equivalent symmetric solutions.
>
> ### Concrete Example (Figure 1 in paper)
> ---
> Suppose two agents must serve tasks A and B. In multi-agent problems, the solution construction trajectory where Agent 1 is assigned to A followed by Agent 2 assigned to B yields the identical solution to the reverse assignment order (assigning Agent 2 to B first, then Agent 1 to A).
> Standard self-improvement (SI) supervises on a single "best" sequence (e.g., Agent 1 $\to$ A first), treating the remaining symmetric choices as errors. Consequently, during training:
> - The chosen action (Agent 1 $\to$ A) is reinforced as the unique correct target
> - The valid symmetric alternative (Agent 2 $\to$ B) is penalized via the softmax denominator because it competes for probability mass at that specific time step
> - The policy effectively learns to avoid this equally valid start to the solution
>
> While the stochasticity might help the policy *explore* symmetric actions during generation, it receives **contradictory supervision**: one symmetric action is labeled correct while others are implicitly labeled incorrect through the cross-entropy loss gradient (see Eq. 37 in Appendix E.1).
>
> Moreover, augmenting the dataset to include all symmetric orderings is non-trivial: after one assignment is made, the state transitions diverge, so “rewinding” to enumerate equivalent joint assignments requires reconstructing counterfactual states, which is infeasible in problems with inter-agent dependencies like precedence constraints (e.g., scheduling).
>
> ### MACSIM's Solution
> ---
> Ultimately, our policy generates actions for all agents under the current state $s_t$ and MACSIM’s multi-action imitation objective treats the entire matching $\mathbf{a}_t^\ast$ (i.e., the set of all agent–task assignments at state $s_t$) as the supervision target.
>
> Our set-based loss (Eq. 6):
>
> $$ \mathcal{L} = -\sum_{i=1}^{M} \text{log} \, P(v_i | m_i)$$
>
> has two key properties:
>
> 1. Permutation invariance: all agent–task pairs are supervised jointly; no arbitrary ordering is imposed.
>
> 2. No penalization of symmetric alternatives: each agent–task pair is an independent positive example, avoiding cross-entropy conflicts inherent to sequential next-token training.
>
> As a result, the policy can learn to assign probability mass to multiple symmetric actions without being penalized for doing so.
>
> ### Empirical Evidence
> ---
> Table 4 demonstrates this effect directly: models trained with our set-based loss ($\mathcal{L}_\text{CE}$) consistently outperform those trained with single-action supervision, with the gap widening as the number of agents increases (Figure 3b), precisely when symmetries become more prevalent.
>
> We hope this clarifies the fundamental distinction and the motivation behind our design. We appreciate the opportunity to elaborate on this point and are happy to provide further explanation if needed.
>
> Sincerely,
> MACSIM authors

---

### Official Review · Reviewer_t8zx · 2025-10-31

**Soundness:** 3
**Presentation:** 2
**Contribution:** 2
**Rating:** 4
**Confidence:** 3

**Summary:**

This paper identifies a critical limitation in existing self-improvement (SI) methods, such as those proposed by Corsini et al. (2024) and Pirnay & Grimm (2024). The authors argue that the standard "next-token" prediction paradigm, which learns from a single expert trajectory, fails to account for agent-permutation symmetries in multi-agent problems. This forces the model to learn an arbitrary agent order, hindering coordination and sample efficiency. To solve this, the paper proposes MACSIM (Multi-ACtion Self-Improvement). The core contribution is a framework that: 1) Uses a policy to predict a single, joint logit matrix ($M \times N$, a "$heatmap$" for all agents and tasks in one forward pass. 2) Employs a set-prediction loss ($\mathcal{L}_{CE}$) to supervise this heatmap on the set of expert assignments, making the training signal invariant to agent permutation. The authors claim MACSIM achieves higher solution quality, better sample efficiency, and "drastically accelerated" solution generation on multi-agent scheduling and routing problems.

**Strengths:**

1. The paper’s primary strength is its clear identification of the "agent-permutation symmetry" problem. The diagnosis that standard SI's next-token supervision (SLIM) implicitly punishes valid, symmetric solutions as "errors" is a significant contribution.
2. The decoupling of the (expensive) one-shot policy evaluation to get the joint logits from the (fast) $M$-step autoregressive sampling (Algorithm 1) is a clever way to amortize computation while guaranteeing a conflict-free joint action.
3. The experimental results show that MACSIM consistently outperforms its baseline (SLIM).

**Weaknesses:**

1. The paper's core claim is that it better handles multi-agent symmetries. The authors cite DPN (Zheng et al., 2024) in their related work, noting it also targets multi-agent permutation symmetries, albeit using a different method (symmetric baselines). However, despite both MACSIM and DPN being applied to min-max routing problems (HCVRP), DPN is completely absent from the experimental comparisons in Table 1. DPN is a state-of-the-art method that directly addresses the same core problem (multi-agent symmetry). Failing to compare against it empirically, while instead comparing against a re-implemented SLIM baseline, significantly weakens the paper's claims of superiority in this domain.
2. The paper positions itself as an improvement over standard self-improvement (SI) but only provides a direct experimental comparison against a re-implemented "SLIM" (Corsini et al., 2024). It fails to provide a direct experimental comparison to other significant and recent work in SI, notably Pirnay & Grimm (2024), which it cites alongside SLIM as the methods it aims to improve upon. While the authors group these methods together conceptually, they are not identical. A direct comparison against Pirnay & Grimm (2024) would be necessary to fully substantiate the claim of superiority over the class of standard SI methods.

**Questions:**

1. Your paper cites DPN (Zheng et al., 2024) and notes that it also addresses agent-permutation symmetries. Given that both your method and DPN are evaluated on min-max routing problems (HCVRP), why was DPN not included as a direct experimental baseline in your empirical comparison (Table 1)?
2. Could you provide a computational profile of your model at inference? Specifically, what percentage of the wall-clock time is spent on the $T$-step policy re-encodings versus the $M \times T$-step autoregressive sampling (Algorithm 1)? This would clarify if the accelerated part is a meaningful portion of the total inference time.
3. You train with the $\mathcal{L}_{CE}$ loss, which treats all $M$ agent assignments as independent classification problems. However, you infer using a sequential, dependent sampling algorithm (Algorithm 1). Why is this independent loss sufficient for learning a policy that must produce coordinated, dependent assignments?

---

> ### Author Response · Authors · 2025-11-20
>
> Dear reviewer t8zx,
>
> Thank you for the detailed and constructive review. We address each point below.
>
>
> ### W1 / Q1 — Missing DPN Baseline
> ---
> DPN (Zheng et al., 2024) was initially omitted because the official codebase does not implement HCVRP. For the rebuttal, we implemented DPN for HCVRP based on the released code and added it to Table 1 (see revised manuscript). The implementation is available in our anonymous repository.
>
> Even after adding DPN, **MACSIM remains the strongest neural baseline on HCVRP**. DPN also fails to outperform $\text{DRL}_{LI}$, SLIM, and 2D-Ptr. This is consistent with prior findings (e.g., Liu et al., 2024): sequential agent-wise construction methods like DPN struggle on capacitated VRP, where each agent must generate multiple tours. Methods that construct routes for all agents jointly (MACSIM, SLIM, 2D-Ptr) consistently perform better due to enhanced coordination.
>
> ### W2 — Missing Pirnay & Grimm (Gumbeldore) Baseline
> ---
> Gumbeldore introduces a sophisticated **sampling** mechanism (stochastic beam search (SBS) with Gumbel-Top-k), but the **self-improvement (SI) algorithm is identical to SLIM**:
>
> 1. Sample multiple trajectories.
> 2. Pick the best under the task objective.
> 3. Train with standard next-token cross-entropy.
>
> Thus, Gumbeldore is a **sampling enhancement**, not a different SI method. We did not include it for three reasons:
>
> - **Orthogonality:** Our contributions address *structural* limitations of single-action SI via multi-agent policy + set-based loss. These improvements are independent of any sampling method and are improvements *to the SI algorithm itself*. SBS can be combined with MACSIM, but that is outside this paper’s scope.
> - **Cost:** SBS is computationally heavy. A main contribution of MACSIM is reducing training/inference latency; adding SBS would undermine this.
> - **Exploration:** SBS mainly combats limited solution diversity and as such training stagnation. We employ entropy regularization and never run into such issues.
>
> In summary, Gumbeldore does not provide a stronger SI baseline than SLIM; its gains stem from expensive sampling, not a fundamentally different learning rule. Including SBS would not change our conclusions.
>
>
>
> ### Q2 — Inference Time Profiling
> ---
> We profiled inference latency on FFSP ($N=50$, $S=3$, $M_i \in \{4,6,8\}$) and decomposed wall-clock time into (i) policy evaluation and (ii) sampling.
>
> **Baseline (SLIM).**
> Stepwise calls to the policy dominate inference (96–99% of total time). In FFSP 50×3×4, policy evaluation takes 3100 ms vs. 120 ms for sampling.
>
> **MACSIM.**
> By amortizing computation of a single forward pass over $M$ agents , MACSIM reduces policy evaluation by **12–16×** (e.g., 234 ms vs. 3100 ms). Sampling time stays similar (125 ms vs 120ms), but its share of total time increases because the policy component becomes much cheaper.
>
> These results (new Table 8, App. D.6) confirm that MACSIM accelerates the dominant bottleneck without adding sampling overhead.
>
>
>
> ### Q3 — Sufficiency of the Independent Loss
> ---
> Despite being per-agent, the cross-entropy loss $\mathcal{L}_{CE}$ is sufficient for coordination through three mechanisms. We added Appendix G with new experiments and visualizations.
>
> ### 1. Architectural Coupling
> All agents share model parameters; cross-attention fuses information across agents/tasks. Although logits are decoupled in the loss, gradients w.r.t. shared $\theta$ force the encoder to shape representations so agents specialize on different tasks.
>
> ### 2. Implicit Constraint Learning
> Training on valid matchings induces a gradient field that suppresses conflicting assignments. For an expert pair $(m, v^\ast_m)$,
> the set-based loss increases $L_{m,v^\ast_m}$ and decreases $L_{m,v}$ for all $v \neq v^\ast_m\$. Importantly, tasks assigned to other agents fall into this negative set, naturally discouraging conflicts - unlike global softmax losses that penalize correct future assignments of other agents.
>
> ### 3. Inference-Time Coupling
> Algorithm 1 samples without replacement from the flattened logits. High-confidence agent–task pairs are likely sampled early, causing immediate masking of selected tasks. This converts confidence into **priority**, enforcing coordination during decoding.
>
> ### Empirical Evidence
> - Table 3: replacing logit-based sampling order with random/fixed order degrades performance, especially with more agents.
> - Appendix G: conflict rate under argmax decreases throughout training; heatmaps show clear separation of agent–task preferences.
>
> **Conclusion:**
> $\mathcal{L}_{CE}$ leverages structured expert data + shared architecture to learn conflict-free representations, while the autoregressive sampler enforces hard constraints. This combination proves both theoretically sound and empirically effective.
>
> ---
>
> Thank you again for your detailed review. We hope the added baselines, analyses, and experiments address your concerns.
>
> Sincerely,
> MACSIM authors

---

> > ### Comment · Reviewer_t8zx · 2025-11-28
> >
> > Thank you very much for your detailed response, which addresses most of my concerns. I will increase the rating accordingly.

---

> > > ### Author Response · Authors · 2025-11-28
> > >
> > > Thank you very much for your continued engagement and for reconsidering your score. We are glad that our response successfully addressed your concerns, and sincerely appreciate your detailed feedback as it helped us strengthen the positioning of our work.

---

### Official Review · Reviewer_Q1dc · 2025-10-31

**Soundness:** 4
**Presentation:** 3
**Contribution:** 3
**Rating:** 6
**Confidence:** 3

**Summary:**

This paper presents a multi-agent extension of the self-improvement paradigm for neural combinatorial optimization (NCO). Instead of predicting one action per step, the proposed method jointly samples actions for all agents from the full joint-agent action space. By incorporating agent-permutation symmetry into both the policy design and training objective, the approach improves coordination among agents and produces higher-quality solutions compared to existing self-improvement baselines.

**Strengths:**

The paper effectively integrates ideas from reinforcement learning–based NCO and self-improvement learning. It makes good use of the MatNet architecture, originally developed for RL, and adapts it to the self-improvement setting. The main idea is simple yet powerful—using the joint action score matrix to derive agent–task matchings, which is both logical and effective. In addition, the paper is very well written, with rigorous equations and precise mathematical definitions.

**Weaknesses:**

While the paper presents a clear and elegant formulation, the novelty appears somewhat incremental, as it mainly builds upon existing self-improvement methods by reformulating them for multi-agent joint-action modeling.

The method relies on full re-encoding of the problem state at each step, which may hinder scalability to larger instances. Moreover, since each re-encoding operates only on the updated state—without explicitly retaining information about past actions or the original problem context—the approach may have limited applicability to problems with temporal dependencies or long-horizon constraints.

**Questions:**

None

---

> ### Author Response · Authors · 2025-11-20
>
> Dear reviewer Q1dc,
>
> We sincerely thank you for the thoughtful feedback and appreciate your positive assessment of our formulation, rigor, and integration of self-improvement (SI) with multi-agent modeling. We address your concerns about (1) novelty, (2) scalability, and (3) temporal dependencies below.
>
> ### 1.) On Novelty and Contribution
> ---
> We respectfully disagree that our contribution is incremental. MACSIM introduces **three fundamental innovations**:
>
> 1. **Joint, conflict-free multi-agent action generation**: We introduce the first joint-action generation mechanism that produces conflict-free multi-agent assignments in a single forward pass. Unlike PARCO, where independent per-agent predictions require heuristic, non-differentiable conflict resolution, MACSIM integrates coordination directly into the sampling distribution through sequential normalization over feasible pairs, ensuring coordinated, conflict-free actions inherently.
>
> 2. **Permutation-Invariant Set-Based Loss**: Existing SI methods operate on single-action labels and assume a unique optimal next action. This assumption fundamentally breaks in multi-agent CO, where large equivalence classes of agent–task assignments exist. Our method replaces this by learning via a set-based loss from all pseudo-expert agent-task assignments observed for state $s_t$. Our theoretical analysis (Section E, Theorem 2) justifies why the set-based cross-entropy loss is a sound surrogate for the intractable permutation-invariant objective. Empirically we demonstrate (Table 4) that it significantly outperforms alternative losses including single next-token loss as well as direct MLE and Plackett-Luce formulations.
>
> 3. **Skip Token with Annealed Penalty**: This design enables agents to learn *when* deferring decisions is beneficial while maintaining construction efficiency -- a critical capability for problems with strong inter-agent dependencies as demonstrated through our ablations (Figure 5, Table 3).
>
> **Empirical Impact**: Together, these innovations yield consistent SOTA across three domains (Tables 1,2,5,6), with gaps widening as agents increase (Figure 3b). For FFSP-50×8×3, MACSIM reduces makespan by ~6% vs. standard SI, **10× faster inference** (Figure 3c), and **nearly 10× faster training** with superior stability (Figure 6).
>
> ### 2.) On Scalability and State Re-Encoding
> ---
> We appreciate your concerns about scalability. We provide both theoretical justification and extensive new empirical evidence [in our revised manuscript](https://openreview.net/pdf?id=6KrETIaOYD) demonstrating that our design choices actually provide *fundamental* scalability advantages. In summary:
>
> **Memory Efficiency vs. RL-Based NCO**: RL baselines (e.g., MatNet) backpropagate through entire trajectories, while MACSIM trains on single state–action pairs. In new FFSP experiments (Appendix D.5, Table 7), MatNet’s memory grows 1→13 GB (N=100→500), while MACSIM only 0.7→2 GB. This enables larger batches and yields a 4× training-time reduction (264→63 min/epoch at N=500).
>
> **Computational Efficiency vs. Standard SI**: Both MACSIM and SLIM re-encode states stepwise, but MACSIM amortizes this cost over $M$ agent actions per encoding step. This drastically reduces rollout length and yields up to 10× faster training (Fig. 6).
>
> **Future Work**: We acknowledge stepwise re-encoding limitations in latency-critical regimes and propose incremental updates and cached embeddings (Limitations section).
>
> ### 3.) Addressing Temporal Dependencies
> ---
> Regarding your concern about temporal dependencies: our formulation captures temporal structure through the state representation itself. In scheduling problems (FJSP, FFSP), the state explicitly encodes job precedence constraints, machine availability, and partial schedules, which the policy's attention mechanism accesses at each step. The strong results on FJSP (Tables 1, 2, 5), where operations must follow strict precedence constraints, demonstrate effective handling of such dependencies.
>
>
>
> ### Conclusion
> ---
> We believe MACSIM represents a significant advance in NCO by being the first to systematically exploit agent-permutation symmetries in SI learning through principled joint-action modeling and theoretically grounded training objectives. Our new scalability analysis (Appendix D.5) demonstrates that our design choices provide fundamental advantages over both traditional RL-based NCO (memory efficiency enabling large batch sizes) and standard SI (amortized re-encoding reducing training time by factors scaling with agent count). The consistent empirical improvements across diverse problem domains and scales, combined with substantial efficiency gains, demonstrate both the theoretical soundness and practical value of our contributions.
>
> We hope this response addresses your concerns and clarifies the scope and impact of our work. We are happy to provide additional clarification or experiments if needed.
>
> Sincerely,
> MACSIM authors

---

### Author Response · Authors · 2025-12-02
**Discussion Period Summary**

Dear Area Chair and Reviewers,

we would like to sincerely thank all reviewers for their time, constructive feedback, and thoughtful engagement throughout the review process. Since the discussion period is coming to an end and since the OpenReview leak prevents any score updates, we would like to take the opportunity to summarize the rebuttal phase.

All reviewers consistently recognized the paper’s solid technical grounding, clear presentation, and the empirical effectiveness of the proposed Multi-Action Self-Improvement (MACSIM) framework. They highlighted the novelty of extending self-improvement to the multi-agent setting, specifically praising our design choices as "clever" (t8zx) and "simple yet powerful" (Q1dc). Furthermore, reviewers z3tB and tkUT underscored the broad applicability of our method, affirming its significance for the Neural Combinatorial Optimization community.

Reviewer t8zx initially raised valuable concerns regarding the missing DPN baseline and computational profiling, which we addressed with new experiments and analyses. Following our rebuttal, reviewer t8zx confirmed that these issues were fully resolved and stated that they would increase their score accordingly, **bringing all reviews into positive territory**.

In summary, the discussion converged on a shared view that the paper makes a meaningful and well-substantiated contribution to Neural Combinatorial Optimization by introducing a principled framework that exploits agent-permutation symmetries in the realm of self improvement, improves coordination, and achieves both higher solution quality and faster inference. We would particularly like to highlight reviewer t8zx’s acknowledgment that identifying how standard self-improvement’s next-token supervision implicitly penalizes valid symmetric solutions represents a **significant and novel contribution**. This recognition underscores the conceptual importance of our work in addressing a fundamental limitation of existing Neural Combinatorial Optimization methods.

We appreciate the reviewers’ recognition of these contributions and their constructive feedback, which has helped us refine and clarify the final manuscript.

We thank all reviewers and the Area Chair once again for their time and thoughtful consideration.

Sincerely,

MACSIM Authors

---

### Meta-Review · Area_Chair_tUY3 · 2026-01-05

**Summary:**

This paper presents a multi-agent extension of the self-improvement paradigm for neural combinatorial optimization (NCO). Instead of predicting one action per step, the proposed method jointly samples actions for all agents from the full joint-agent action space. By incorporating agent-permutation symmetry into both the policy design and training objective, the approach improves coordination among agents and produces higher-quality solutions compared to existing self-improvement baselines.

Generally, this paper is well-written and may contribute to the direction. I tend to accept this paper as a poster.

**Reviewer Concerns:**

Reviewers' concerns have been generally addressed.

**Reviewer Scores:**

Reviewer t8zx has committed to improving the assessment.

---

### Decision · Program_Chairs · 2026-01-26

Accept (Poster)